# Antibiotic-chemoattractants enhance neutrophil clearance of *Staphylococcus aureus*

Jennifer A. E. Payne [1,2,3 ✉], Julien Tailhades[1,2,3], Felix Ellett [4], Xenia Kostoulias [5], Alex J. Fulcher [6], Ting Fu[7], Ryan Leung[1], Stephanie Louch [1], Amy Tran [1], Severin A. Weber[1,3], Ralf B. Schittenhelm [8], Graham J. Lieschke [9], Chengxue Helena Qin [7,10], Daniel Irima [4], Anton Y. Peleg [5,11,12] & Max J. Cryle [1,2,3 ✉]

The pathogen *Staphylococcus aureus* can readily develop antibiotic resistance and evade the human immune system, which is associated with reduced levels of neutrophil recruitment. Here, we present a class of antibacterial peptides with potential to act both as antibiotics and as neutrophil chemoattractants. The compounds, which we term 'antibiotic-chemoattractants', consist of a formylated peptide (known to act as chemoattractant for neutrophil recruitment) that is covalently linked to the antibiotic vancomycin (known to bind to the bacterial cell wall). We use a combination of in vitro assays, cellular assays, infection-on-a-chip and in vivo mouse models to show that the compounds improve the recruitment, engulfment and killing of *S. aureus* by neutrophils. Furthermore, optimizing the formyl peptide sequence can enhance neutrophil activity through differential activation of formyl peptide receptors. Thus, we propose antibiotic-chemoattractants as an alternate approach for antibiotic development.

[1] Infection and Immunity Program, Monash Biomedicine Discovery Institute, Department of Biochemistry and Molecular Biology, Monash University, Clayton, Victoria 3800, Australia. [2] ARC Centre of Excellence for Innovations in Peptide and Protein Science, Monash University, Clayton, Victoria 3800, Australia. [3] EMBL Australia, Monash University, Clayton, Victoria 3800, Australia. [4] BioMEMS Resource Center, Center for Engineering in Medicine and Surgical Services, Massachusetts General Hospital, Shriners Hospital for Children, and Harvard Medical School, Charlestown, MA 02129, USA. [5] Infection and Immunity Program, Monash Biomedicine Discovery Institute and Department of Microbiology, Monash University, Clayton, Victoria 3800, Australia. [6] Monash Micro Imaging, Monash University, Clayton, Victoria 3800, Australia. [7] Drug Discovery Biology, Monash Institute of Pharmaceutical Sciences, Monash University, Parkville, Victoria 3052, Australia. [8] Monash Proteomics and Metabolomics Facility, Monash University, Clayton, Victoria 3800, Australia. [9] Australian Regenerative Medicine Institute, Monash University, Clayton, Victoria 3800, Australia. [10] Baker Heart and Diabetes Institute, Melbourne, Victoria 3004, Australia. [11] Department of Infectious Diseases, The Alfred Hospital, Melbourne, Victoria 3004, Australia. [12] Central Clinical School, Monash University, Melbourne, Victoria 3004, Australia. ✉email: jennifer.payne@monash.edu; max.cryle@monash.edu

Antibiotics have become a cornerstone of modern medicine, allowing the prevention and treatment of once deadly infections. However, their durability is hampered by the emergence of bacterial resistance[1,2]. This is particularly notable for the Gram-positive bacteria *Staphylococcus aureus* where the rise of methicillin resistance (MRSA) has led to an increased reliance on last-resort antibiotics. MRSA is widespread, accounting for 37% of *S. aureus* bloodstream infections and also has higher mortality rates[3]. In addition to developing antibiotic resistance, *S. aureus* also employs an arsenal of secreted factors to evade the host immune system[4]. Avoiding detection by cells of our innate immune response ensures that first-line responders, such as neutrophils cannot come into play to protect us from this infection[4]. Additionally, resistance to clinical antibiotics, such as methicillin, in *S. aureus* is leading to the emergence of strains with greater immune evasion[5,6]. The increased difficulty in eradicating these strains, therefore, leads to persistent and deadly infections[6]. Understanding the critical need for the development of new antibiotics, the World Health Organization has listed MRSA as a high priority target.

New approaches to tackle this problem are urgently required, which do not rely on the increasingly risky strategy of identifying new antibiotics from nature (the source of the vast majority of clinical agents[7]) or increasing the toxicity of the antibiotic. Designer immunotherapeutics to target bacterial infections are one example of a new strategy that is being explored to fill this innovation gap[8,9]. Our body's own immune system provides inspiration and is central to the efficacy of such strategies, with one example being self-assembling peptide nanonets that entrap bacteria and kill them with embedded antimicrobials, taking their inspiration from neutrophil extracellular traps[10]. Another approach is mimicking the body's own host defense peptides that combine two functions in one—direct antimicrobial activity and immune modulation or enhancement[11,12]. The ability of antibiotic-resistant bacteria to avoid immune detection can be overcome using these dual-acting molecules by enhancing the chemoattractant gradient around the bacteria. The establishment of this gradient is an essential step for the recruitment of innate immune cells, such as neutrophils, for prevention, and resolution of infection.

One of the main pathogen-associated chemoattractants are formylated peptides (fPeps). These fPeps are short peptides (3–5 amino acids) that activate innate immune cells by binding to formyl peptide receptors (FPR), stimulating chemotaxis and phagocytosis of the bacteria[13,14]. The importance of fPeps and FPR activation for preventing infection is highlighted by the diversity of proteins that *S. aureus* produces to counter both chemotaxis and FPR activation[4]. These fPeps are the N-terminal fragments of bacterial proteins, as formyl methionine initiates protein biosynthesis in bacteria[13,15]. These bacterially derived peptides are strong inducers of chemotaxis and are necessary for neutrophils to mount a successful immune response; host-derived chemoattractants alone often insufficient for this task[16,17]. This chemoattractant gradient can be enhanced by causing the bacteria to generate more fPeps, as seen in the mechanism of action of actinion, which inhibits bacterial peptide deformylase leading to enhanced release of fPeps from the bacteria[18]. Another strategy being pursued is through artificially enhancing the chemotactic gradient by targeting and attaching chemoattractants, like fPeps, directly to the bacteria. Treatment with an isolated chemotactic agent would be insufficient to generate the gradient required for neutrophil chemotaxis. To provide specificity of immune activation toward the infecting bacteria, a chemotactic agent combined with a bacterial targeting component is necessary.

In the current study, we propose an approach to exploit the properties of fPep chemoattractants by linking them to the clinical antibiotic vancomycin[19,20], thereby producing an immunotherapeutic to treat *S. aureus* infections (Fig. 1). For these vancomycin-fPep conjugates, vancomycin acts as a targeting agent that binds to the *S. aureus* cell wall. The attached fPep then generates a chemoattractant gradient surrounding *S. aureus* cells, improving the recruitment of neutrophils to the site of infection and enhancing phagocytosis of the bacteria[14]. Aspects of linking a fPep to a targeting element have been examined for Gram-negative bacteria using polymyxins[21] and for fungi through use of caspofungin or amphotericin B[22]. However, these studies did not optimize the fPep sequence, which we have found to be essential in order to establishing enhanced activity in these conjugates. The creation of these proof-of-principle antibiotic-chemoattractants required the development of an analytical platform to assess these dual-functional antimicrobial agents that extend beyond the standard assessment of antibiotic action. Through the use of infection-on-a-chip technology, we directly demonstrated that our conjugates modulate the interactions between *S. aureus* and neutrophils by enhancing neutrophil recruitment, phagocytosis, and eventual clearance of *S. aureus* cells. This also proved to be an effective strategy in vivo, where antibiotic-chemoattractants demonstrated reduced bacterial load and improved lung pathology in a mouse MRSA pneumonia model. Taken together, our work demonstrates the promise of possible antibiotic therapy resulting from the targeted activation of the human immune system toward microbes.

## Results

**The antibiotic-chemoattractant binds to the surface of antibiotic-resistant *S. aureus*.** Vancomycin was chosen as the optimal candidate for our antibiotic-chemoattractant approach, as its mechanism of action involves binding to the bacterial surface[19,20]. An initial fPep (**B1**, fMLFK(BODIPY)-Pra-NH$_2$) was generated by reacting the fPep (**R11**) with the commercially available BODIPY-OSu ester. Click chemistry was used to generate the C-terminal BODIPY-labelled vancomycin (**B2**) and the C-terminal BODIPY-labelled fPep-vancomycin conjugate (**B3**)[23,24]. Using super-resolution confocal microscopy, binding of the labelled antibiotic-fPep conjugate was observed to have a similar binding distribution as the labelled vancomycin alone to MSSA and MRSA strains. This correlates with the findings of previous studies investigating labelled glycopeptide antibiotic (GPA) binding[24–26], with binding to the septum in *S. aureus* also similar for vancomycin and the conjugate (Supplementary Fig. 1). Conjugate binding was also observed to *B. subtilis*, suggesting the potential for antibiotic-chemoattracts to be used against Gram-positive bacteria more generally; no binding was observed to *E. coli* (**B1**, Supplementary Fig. 2). Despite binding that followed the same general trends as vancomycin, the antibiotic-chemoattractant displayed a 5–7-fold increase in the minimal inhibitory concentration against 9 clinical strains of hospital-acquired MRSA and MSSA (Supplementary Table 1). Given this, we next explored the optimal attachment site on vancomycin for the chemoattractant cargo in order to ameliorate this reduction in activity.

**Determining the optimum site and linker length for conjugation to vancomycin.** We investigated three possible sites of attachment to vancomycin to minimize the reduction in antimicrobial activity: the primary amine on the sugar (V linked); the N-methylated amine (N-linked); or the carboxyl group (C-linked, Fig. 1). We also explored altering the linker length between the fPep and vancomycin by varying the number of flexible and polar polyethylene glycol (PEG) spacer units (0, 3, or 6 repeats). These fPeps with linkers were prepared by solid-phase peptide synthesis.

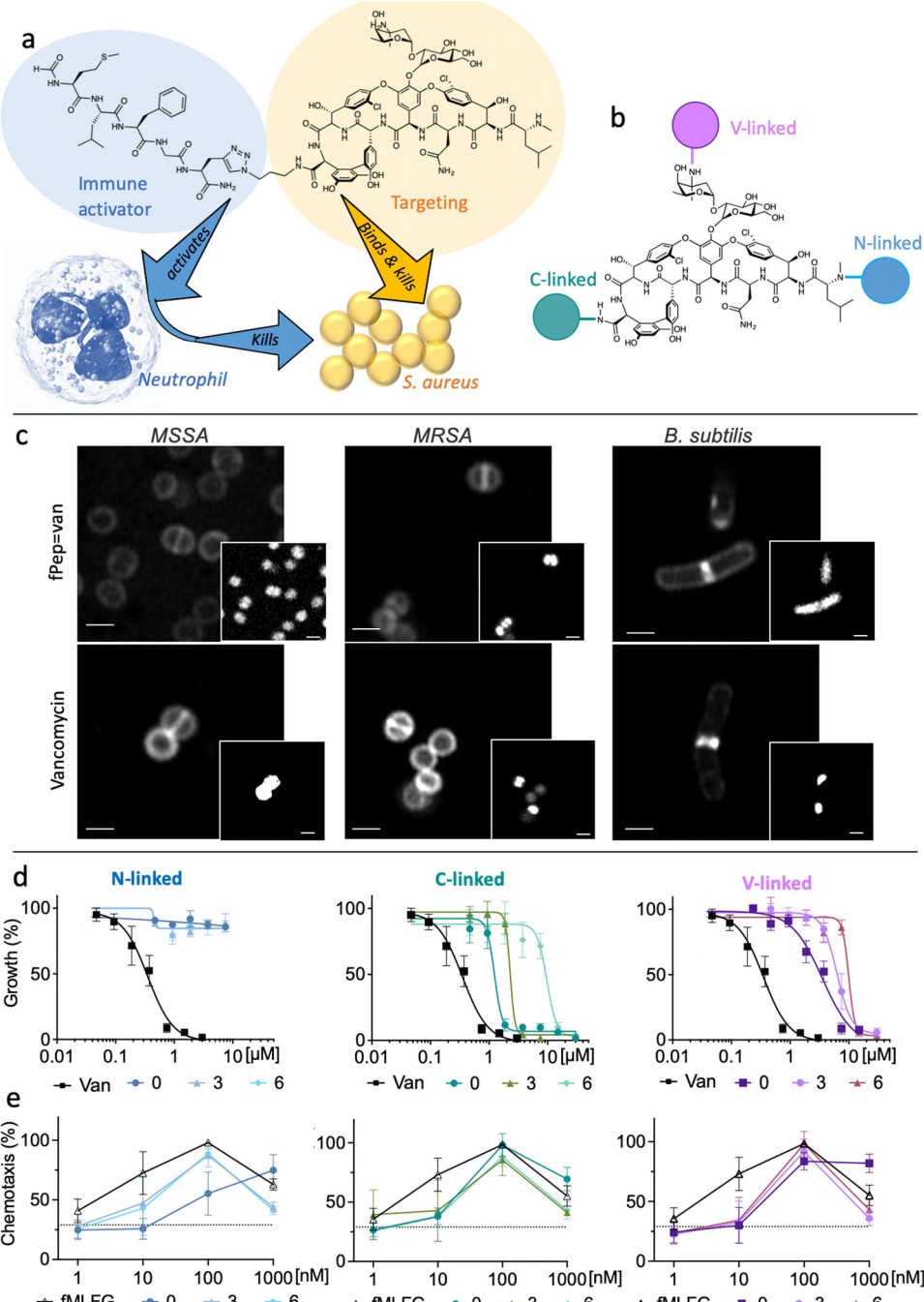

**Fig. 1 Overcoming *S. aureus* immune evasion with immunotherapeutic antibiotics that bind to bacteria, retains activity against bacteria and can recruit neutrophils. a** Our strategy combines the possibility of direct killing of the bacteria with enhancing the neutrophil response. Vancomycin (yellow) targets and binds directly to *S. aureus* cell wall, while the fPep helps the immune response (blue) by enhancing the chemotactic gradient, directing immune cells to the bacterial invader and improving phagocytosis. Neutrophil image modified from[70]. **b** Linkage to vancomycin can be made through three main sites: the vancosamine primary amine (V linked, purple); methylated amine (N-linked, blue); or the carboxyl group (C-linked, green). **c** Fluorescently labelled BODIPY vancomycin (**B2**) or the conjugate (fPep = van, **B3**) imaged by Airyscan super-resolution microscopy localized to the wall of the Gram-positive bacteria *B. subtilis*, *S. aureus*-MSSA, and an MRSA clinical isolate. Scale bars are 1 μm; Insert image is nuclear staining by Hoechst 33342, images are representative of 3 biological replicates. The linkage site to vancomycin and the length of the PEG linker was critical to maintain antimicrobial activity against *S. aureus* as seen in a microbroth dilution assay (**d**) but had little effect on the chemotaxis of human neutrophils in transwell assay (**e**). Three different length PEG linkers were trialed: no linker (0), 3 PEG units (3), or 6 PEG units (6) at the three attachment sites on vancomycin, leading to three different C-linked (**C1**–**3**), N-linked (**N4**–**6**), and V-linked (**V7**–**9**) compounds. The growth of a clinical isolate of *S. aureus*—MRSA was determined relative to the no protein control. Chemotaxis was calculated relative to the no protein control and 100% chemotaxis set as the neutrophil recruitment observed to fMLFG (**FP1**) at 100 nM for each donor. Error bars are SEM, $n = 3$ biologically independent experiments.

Incorporation of a C-terminal L-propargylglycine residue (containing an alkyne group) allowed for the simple conjugation of the fPep to the three different azide-modified vancomycin scaffolds (**GPA2**, **3,** and **4**) using copper-catalyzed click-chemistry[10,27]. Monitoring neutrophil chemotaxis via transwell assays demonstrated that the attachment site and length of the linker did not considerably affect human neutrophil recruitment, with the typical bell-shaped profile for chemotaxis agonists noted for all conjugates. The highest levels of recruitment were observed at 100 nM for all attachment sites and linker lengths, similar to the recruitment exhibited by free fMLFG (**FP1**, Fig. 1). However, at 10 nM there was a decrease in recruitment for all antibiotic-chemoattractants compared to the fMLFG (**FP1**) alone, indicating possible alterations to the fPep chemotaxis profile due to conjugation to vancomycin.

In contrast to neutrophil recruitment, the site of attachment and a minimal linker length was key to maintaining direct growth inhibition of *S. aureus* strains (Fig. 1), which was important to assess as a proxy for bacterial binding. Activity was tested in a microbroth dilution assay against an MRSA strain (A8090). N-linked compounds (**N4–6**) were unable to inhibit MRSA growth at the highest concentration tested (30 µM). This is unsurprising, as N-terminal structural modifications of vancomycin often hinder the essential interactions with the D-Ala-D-Ala motif in the cell wall that are required for antimicrobial activity[23,28]. A loss of antimicrobial activity was also observed for the V-linked compound (**V7**), which increased with linker length (**V8–9**). This loss is possibly explained by the bulky attachment altering vancomycin dimerization that occurs through these sugar groups and that again is important for antimicrobial activity[29,30]. The C-linked conjugate with no spacer (**C1**) retained the highest antimicrobial activity of any of the antibiotic-chemoattractants, albeit only half the activity of vancomycin. Similar to the V-linked conjugates, there was a loss of antimicrobial activity with the inclusion of PEG spacers (**C2–3**).

This reduction in antimicrobial activity would normally suggest that these would not be an effective treatment strategy. However, the key outcome here was that the vancomycin core still bound to the cell wall of *S. aureus* independently of the antibiotic resistance state. Given the antimicrobial activity profiles of all conjugates, fPep attachment through C-linkage with no PEG linker was chosen for further analysis to maximize the potential for bacterial binding. The addition of chemical moieties to the C-terminus of vancomycin has previously been seen to be an effective strategy to enhance antimicrobial activity[23,31–35]. This decision also considered the current drug approval environment that favors the assessment of direct antimicrobial activity for such compound classes.

**Determining the effect of fPep sequence on neutrophil recruitment**. Having established vancomycin as a viable chassis for antibiotic-chemoattractant conjugates through C-terminal attachment, we next turned to optimization of the fPep component and investigated the effect of different fPep sequences on neutrophil recruitment. The sequence of fPeps are important for both potency and differential activation of the two main formyl peptide receptors (FPRs) of neutrophils, FPR1 and FPR2[13]. Whilst high-resolution structures of FPRs are now available[36,37] most of what is known regarding the diversity of sequences recognized by specific FPRs has been defined by fPep libraries[38–40]. However, the influence of fPep sequence on downstream chemotaxis and phagocytosis remains largely unknown[14,41]. We therefore investigated the effect of altering fPep sequence on neutrophil recruitment.

The fMLFG (**FP1**) sequence is a potent chemoattractant, resulting in peak neutrophil chemotaxis at 100 nM. We postulated that our antibiotic-chemoattractants could require an fPep that would maximally activate neutrophils at different concentrations to **FP1** in order to ensure optimal fPep-induced neutrophil recruitment from our conjugates without detrimental overstimulation[42,43]. To identify the optimum fPep, we generated a peptide library based upon the fMLFG sequence using combinatorial peptide synthesis on SynPhase™ lanterns (Fig. 2a). Each lantern was tagged and common synthetic steps could be achieved following a split/mix strategy to speed the elaboration of the library. For library design, we modified each of the residues in the fMLFG sequence using a variety of proteinogenic and nonproteinogenic amino acids (Fig. 2 & Supplementary Table 2) that covered a range of hydrophobicity and size profiles (Supplementary Fig. 2). Using a transwell assay we examined the recruitment of human neutrophils to these peptides (1–1000 nM) and observed that this library could be classified into five different profiles depending on concentration of peak recruitment (Fig. 2b–f). No obvious trends connected the chemotaxis profile observed to the position of the change, solvent-accessible surface area, or the hydrophobicity of the fPep. This was despite the wide range of hydrophobicity profiles tested (ranging from a Log(P) of −2.5 to 2.5 compared to the original fMLFG sequence) and changes in the solvent-accessible surface area (Supplementary Fig. 3). Oxidation and hydrolysis are two common events that can render these sequences inactive. For this reason, we included norleucine as a replacement for methionine as it is resistant to oxidation, thus producing a longer-lived fPep that could benefit the conjugate[44]. Furthermore, peptides that did not contain the formyl group or had a glutamic acid instead of phenylalanine in the position three of the peptide did not lead to neutrophil recruitment at any concentration tested (Supplementary Fig. 4).

We next investigated if the attachment of these peptides onto vancomycin altered their ability to induce neutrophil chemotaxis. Following the strategy described above, we linked representative peptides displaying each of the five chemotactic profiles onto vancomycin (C-linked) and examined neutrophil recruitment in response to these compounds. The attachment of fPeps to vancomycin led to a general shift in neutrophil recruitment towards higher concentrations. This phenomenon was also observed with conjugation of the archetypal fMLFG sequence, likely due to the interaction of the bulky vancomycin molecule altering neutrophil FPR activation. Interestingly, these changes led to altered profiles for fPeps where the maximal chemotaxis was observed at either 100 or 1000 nM, irrespective of the initial chemotaxis profile for the free fPep. In contrast, fPeps that initially induced peak chemotaxis at 100 and 1000 nM, fMLFP (**FP26**) and fMLFK (**FP21**, Fig. 2e), displayed unchanged profiles once conjugated to vancomycin (**C16–17**). For the fMChaFG peptide (**FP11**), which displayed peak recruitment at 10 and 100 nM (Fig. 2c), attachment to vancomycin abolished recruitment below 10 nM (**C11**). Vancomycin alone or linked to MLFG (**C12**) led to no neutrophil recruitment at any of the concentrations tested (Supplementary Fig. 3).

The peptides fMLFL (**FP19**, Fig. 2d) and fMLYG (**FP12**, Fig. 2f), which recruited low numbers of neutrophils across the concentration range tested, also led to the same behavior when conjugated to vancomycin (**C18–19**). These sequences were chosen for conjugation, as the low levels of neutrophil recruitment made them ideal for testing whether they could improve bacterial killing whilst avoiding potentially deleterious overstimulation of the immune response[42,43]. These conjugates **C18–19**, along with fMLFG = van (**C1**) and fMChaFG = van (**C11**) were all found to have MIC values of 1.8 µM against both MRSA and MSSA strains tested. This indicates that the sequence of the formylated peptide attached to vancomycin does not alter

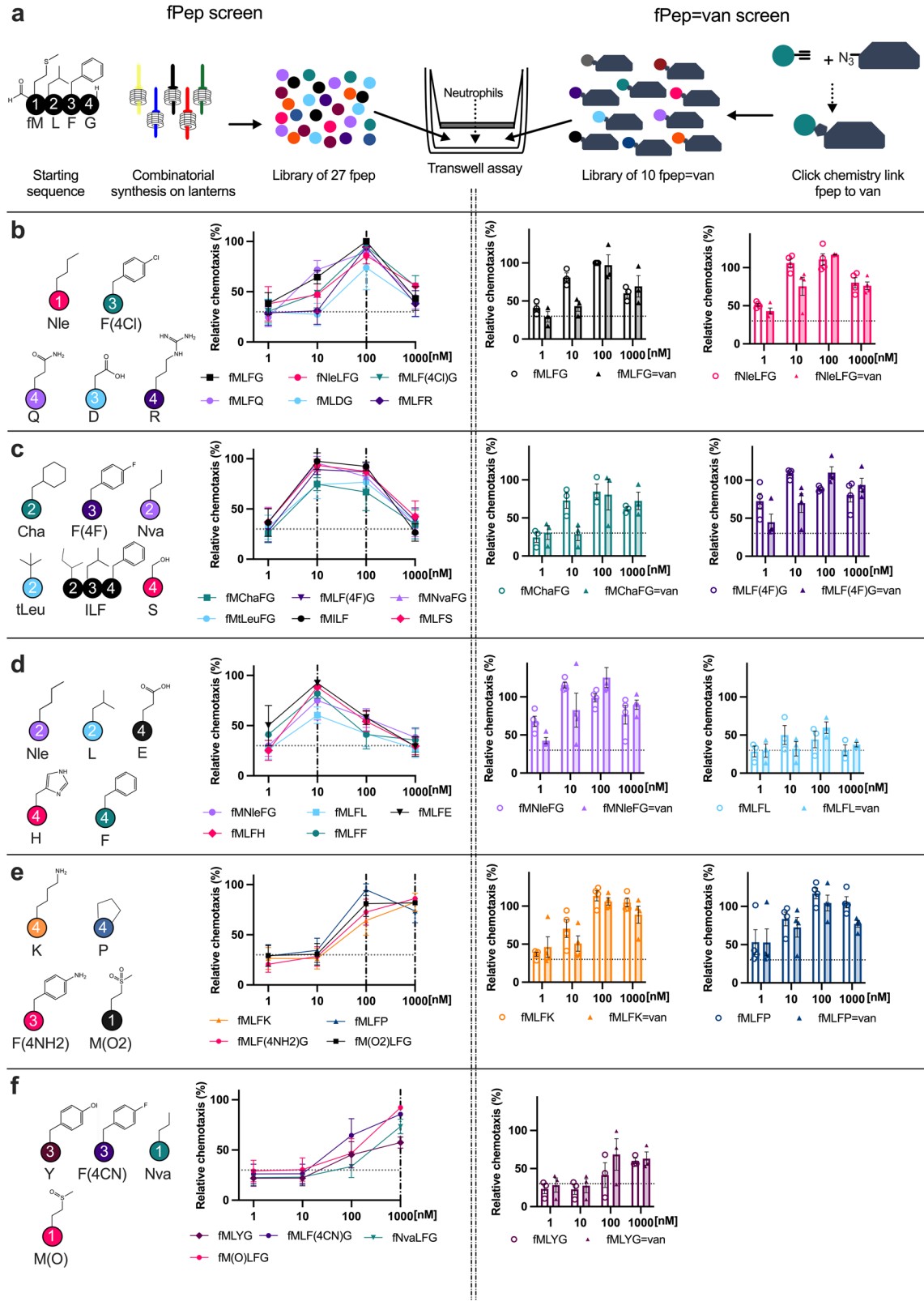

antimicrobial activity, unlike that seen with varying the attachment site of the fPep to vancomycin. It also allows the direct comparison of the neutrophil killing activity of these different conjugates as they possess the same level of direct antimicrobial activity against the bacterial strains tested.

Testing neutrophil recruitment and direct antimicrobial killing separately allowed us to independently optimize the individual activities of our antibiotic chemoattractant. Next, we sought to explore how direct antibiotic activity and binding, combined with induction of neutrophil recruitment, might lead to enhanced killing of *S. aureus*.

**Effects of fPep sequence on neutrophil recruitment and phagocytosis of S. aureus.** Using an infection-on-a-chip

**Fig. 2 The fPep sequence either free or linked to vancomycin affects neutrophil recruitment. a** Schematic depiction of the process to synthesize and test a library of fPeps centered on the fMLFG sequence. The initial library of 27 fPep were generated by combinatorial solid-phase peptide synthesis on SynPhase™ lanterns. A transwell assay was used to determine the chemotaxis of human neutrophils to these peptides in a concentration range of 1–1000 nM. The library of fPeps were grouped into 5 profiles (line graphs, B-F, $n = 3$ biological independent experiments, error bars are SEM) based on the concentration of the fPep that resulted in the greatest recruitment of neutrophils. Peak recruitment was observed at: 100 nM (**b**), both 10 and 100 nM (**c**), 10 nM (**d**), 100 and 1000 nM (**e**), and 1000 nM (**f**). The changes in fMLFG sequence are indicated by the altered residue number (1 = fM, 2 = L, 3 = F, 4 = G), with the side chain of the switched amino acid indicated. Representative fPeps from each of these profiles were linked to the C-terminus of vancomycin via click chemistry and retested for the ability to recruit neutrophils using transwell assay (bar graphs, solid bars represent conjugate). Chemotaxis was calculated relative to the no protein control and 100% chemotaxis set as the neutrophil recruitment observed to fMLFG at 100 nM for each donor, $n = 3$–4 biological independent experiments, error bars are SEM. Dotted line on graphs indicates average recruitment to 0 nM.

microfluidic device, we next quantified neutrophil recruitment, phagocytosis of bacteria through pHrodo fluorescence and growth of GFP expressing *S. aureus* in response to our antibiotic-chemoattractant (Fig. 3a)[45]. Curiously, the kinetics of neutrophil recruitment were slightly faster for the conjugated fMLFG compound (**C1**) compared to the free peptide (**FP1**, Fig. 4), while fMChaFG showed similar kinetic profiles for both the free (**FP11**) and conjugated (**C11**) peptides (Fig. 3). Neutrophil chemotaxis into the microchambers occurred from the start of the assay, while pHrodo fluorescence, indicating the phagocytosis of bacteria, was detected after 60 min (Fig. 3). Importantly, the increased recruitment of neutrophils into the chamber mediated by the antibiotic-chemoattractants resulted in enhanced control of *S. aureus* growth compared to vancomycin or fPep alone (Fig. 3 and Supplementary Fig. 5). The control of *S. aureus* growth here is due to the neutrophil action, rather than the direct antimicrobial activity of the compounds: this was confirmed by the observations that the MLFG = van compound, which does not result in neutrophil recruitment, but that displays similar direct antimicrobial activity as **C1** and **C11**, resulted in no observable growth inhibition of *S. aureus* by neutrophils in the microfluidic assay(Supplementary Fig. 5).

Although there was more rapid recruitment of neutrophils observed for the fMLFG antibiotic-chemoattractant (**C1**), there was only a slight increase in phagocytosis over the free fPep (**FP1**). In contrast, fMChaFG (**FP11**) and its corresponding antibiotic-chemoattractant (**C11**) generated similar neutrophil recruitment into the microchamber. However, the antibiotic-chemoattractant induced more rapid phagocytosis and higher levels of total phagocytosis compared to the free peptide (Fig. 3e). This indicates that the fMChaFG antibiotic-chemoattractant (**C11**) increased the number of bacteria phagocytosed per neutrophil (observed with all neutrophil donors to different extents), which was not observed for the fMLFG conjugate (Fig. 3f). For antibiotic-chemoattractants based on fMLFL (**C18**) and fMLYG (**C19**), enhancement of phagocytosis appeared to be donor specific, although quantification of this effect was limited by their lower levels of neutrophil recruitment (Fig. 3f).

To directly demonstrate that antibiotic-chemoattractants bind to *S. aureus* and result in an fPep gradient that stimulates neutrophil recruitment, we next incubated the fMLFG antibiotic-chemoattractant (**C1**) or the free fPep (fMLFG, **FP1**) with *S. aureus* bioparticles, before washing the samples to the extent where no neutrophil recruitment was observed for the free peptide treated sample (fPep estimated to be less than 0.1 nM). Strikingly, the washed antibiotic-chemoattractant treated *S. aureus* bioparticles induced recruitment of neutrophils to the same extent as 100 nM fPep (Fig. 3g). This demonstrates both initial binding and secondary chemoattractant gradient formation following treatment with these antibiotic-chemoattractants.

**Antibiotic-chemoattractants interact with FPR1 and FPR2.** The sequence of fPeps are known to be important for both potency and

differential activation of the two main FPRs present on neutrophils, FPR1 and FPR2. The fPeps interact with two FPRs expressed on human neutrophils, and although these G-protein coupled receptors share 69% sequence identity, they have distinct preferences for fPep binding. In general, FPR1 recognizes short fPeps with high affinity. In contrast, FPR2 is more of a promiscuous receptor that recognizes a diversity of ligands, including longer fPeps within its larger binding pocket[36,37]. Here we have observed that there is a sequence-specific enhancement phagocytosis by fMChaFG conjugate (**C11**), a downstream result of FPR activation. Such enhanced phagocytosis could well be due to the larger, more hydrophobic sequence selectively activating FPR2, which has a larger binding pocket and accepts more hydrophobic peptides than the FPR1[46]. To examine this, we added the Rhodamine B labelled FPR2 antagonist, PBP10 (RhB-PBP10), to human neutrophils to determine which fPeps and conjugates could compete with this FPR2 antagonist (Fig. 4a). Conjugation of fMLFG (**C1**) resulted in a decrease in RhB-PB10 binding compared to the free fMLFG peptide (**FP1**), suggesting that conjugation resulted in the fMLFG sequence becoming a stronger FPR2 binder, showing similar levels of competition to the known FPR2 preferential binder fMIVIL. This is in agreement with previous observations where other fPep conjugates that changed binding preference from FPR1 to FPR2 due to the increase in size of the peptide[47,48]. The free fMChaFG peptide was able to compete with RhB-PB10 binding, while in contrast the conjugation of the fMChaFG peptide resulted in an inability to compete with RhB-PB10. Here, treatment with the conjugate form (**C11**) was shown to lead to similar levels of RhB-PBP10 binding as the known preferential FPR1 binder fMLF, the negative control.

Binding to FPRs can lead to the activation of a range of downstream pathways. We examined two of these pathways through monitoring the phosphorylation of ERK, and the mobilization of calcium. Using CHO cells overexpressing either human FPR1 or FPR2, the levels of ERK phosphorylation and the calcium mobilization were monitored in response to both fMLFG and fMChaFG in both the free and conjugated forms (**FP1**, **C1** and **FP11**, **C11**), as has been previously reported[49,50]. All peptides resulted in a peak-response time of 5 min for ERK phosphorylation (Fig. 4c). Whilst this activation quickly dissipated in the FPR1 cell line, it appears to be relatively more sustained in the cells expressing FPR2 compared to the fMLF control. β-arrestins regulate GPCR signaling through receptor desensitization and internalization[51], and therefore this sustained FPR2 signaling through ERK may be due to decrease in β-arrestin dependent MAPK kinase (ERK1/2) activation in this pathway compared to the fMLF. The concentration-response curve was also examined at this peak response of 5 min. In general, conjugation of both fPeps resulted in a lower potency for FPR1 activation compared to their free counterparts for both ERK phosphorylation (10-fold increase, Fig. 4d) and calcium mobilization (>90-fold decrease in pEC$_{50}$, Fig. 4b, d) and an increase in potency for FPR2 calcium mobilization (>90-folds increase in pEC$_{50}$, Fig. 4b, e). Despite lacking the ability to outcompete RhB-PB10 in binding to FPR2,

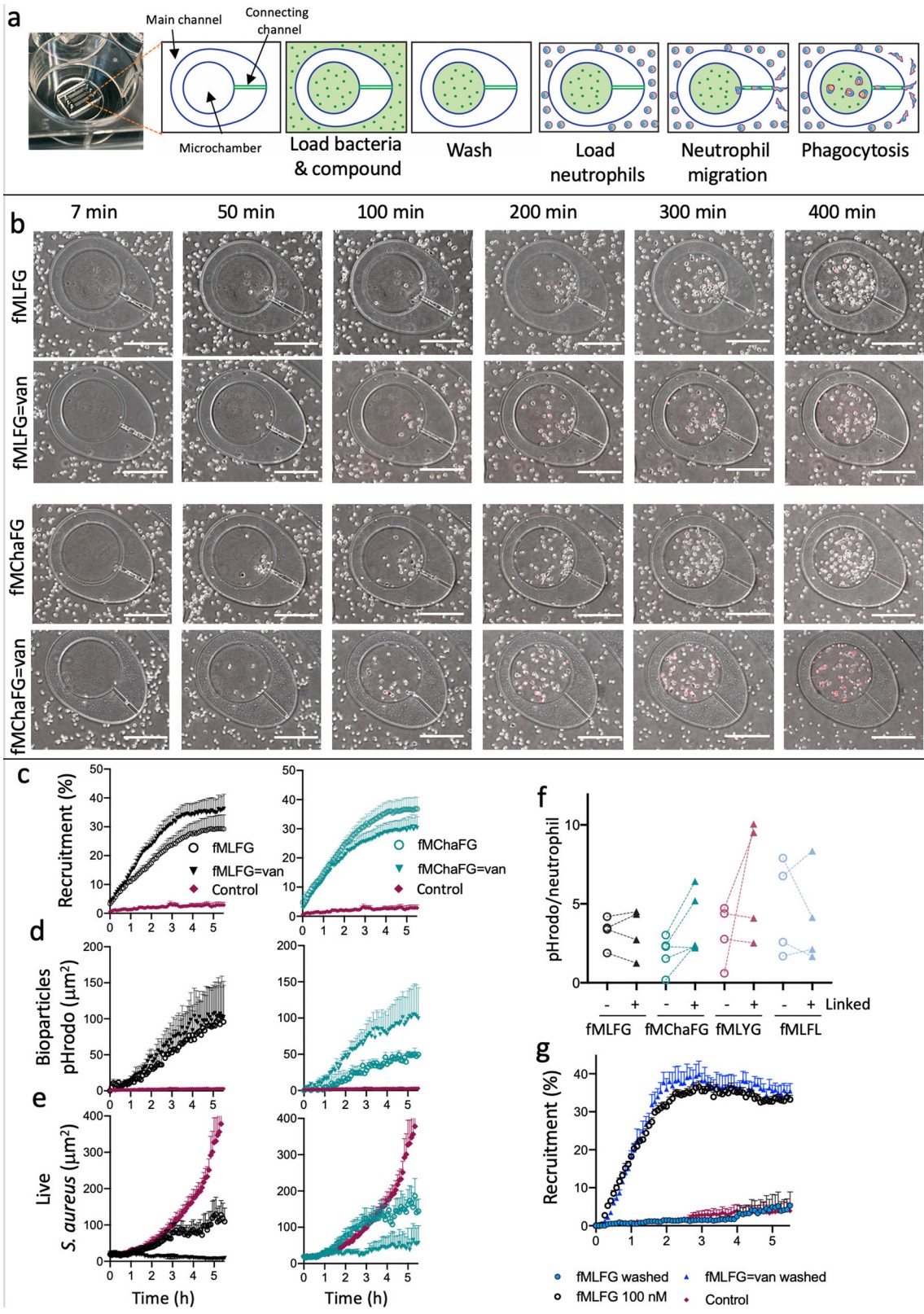

the fMChaFG conjugate (**C11**) demonstrated ERK1/2 phosphorylation and calcium mobilization in FPR2 expressing cells similar to the fMLFG conjugate (**C1**).

**Antibiotic-chemoattractants reduce inflammation and bacterial load in a mouse model of *S. aureus* pneumonia.** Based on the

infection-on-a-chip results, our most promising antibiotic-chemoattractant (fMChaFG = van, **C11**) was tested for efficacy in a MRSA mouse pneumonia model (Fig. 5a). Prior to these experiments, the conjugate **C11** was confirmed to recruit mouse neutrophils using the transwell-based assay; this was tested as FPR sequence specificity differs between mouse and human. The mouse pneumonia model was performed by first establishing an

**Fig. 3 Linking a fPep to vancomycin enhances the phagocytosis activity of neutrophils. a** The infection-on-a-chip device used to monitor neutrophil migration and phagocytosis over time present in a six-well plate. Within this microfluidic device the egg-like microchambers in the main channel were loaded with *S. aureus* plus compound, washed, and neutrophils loaded. A chemotactic gradient of compound establishes from the microchamber along a single cell wide connecting channel. Bacteria are labelled with the pH activated dye, pHrodo to allow monitoring of phagocytosis. **b** Time course of neutrophils migrating into the microchamber containing *S. aureus* bioparticles labelled with pHrodo, in the presence of free fPep (**FP1**, or **FP11**) or conjugated (**C1** or **C11**) at 1000 or 0 nM (control). Images are representative of four donors; scale bars 10 μm. Quantification of neutrophils having migrated into the microchamber (**c**) along with the area of pHrodo fluorescence (**d**), or the area of GFP fluorescence as a measure of *S. aureus* growth (**e**) determined in the presence of conjugated (triangles, **C1** or **C11**) or free fPep (circles, fMLFG **FP1**, black; and fMChaFG **FP11**, teal; at 1000 or 0 nM, control). Data are the average of four donors, with error bars being SEM. **f** The sequence of the conjugate effects efficiency of neutrophil phagocytosis as determined by the area of pHrodo fluorescence per neutrophil recruited into the microchamber at 2 h. Data from each donor is linked by dotted lines. **g** fMLFG conjugated to vancomycin (fMLFG = van **C1**) or free fPep (**FP1**) were incubated with *S. aureus* bioparticles and washed before loading into the microchamber. The washed conjugate treated *S. aureus* recruited similar levels of neutrophils compared to samples with 100 nM fMLFG (open circles). Data are representative of three donors, error bars are SEM.

MRSA infection in the lung, prior to treatment being administered 1 h postinfection intranasally with a dose of 0.2 mg/mouse equivalent of vancomycin. In regard to the area under the concentration-time curve (AUC) of total vancomycin in serum, this dose is 5-fold less than the standard 100 mg/kg equivalent of the clinical dose of vancomycin in humans.

Promisingly, treatment with this low dose of antibiotic-chemoattractant resulted in a 2-fold reduction in MRSA load in the lungs compared to vancomycin alone at 12 hpi (Fig. 5b). Unsurprisingly, histopathology revealed that the lung tissue of the vehicle control mice had the most severe and widespread inflammation and bacterial load, which resulted in dramatic loss of alveoli structure (Fig. 5c, d). In these mice, inflammation was centered on the alveoli, where large numbers of neutrophils, alveolar macrophages, and both extracellular and intracellular bacteria were found. Mice in the vancomycin control treatment group showed reduced severity of inflammation and lower bacterial load, although lung damage was still observed. The fPep and antibiotic-chemoattractant treatment groups both showed reduced inflammation and bacterial numbers compared to vancomycin treatment, as the inflammation did not extend into the lumen of the bronchioles and bronchi (Fig. 5e). Importantly, the antibiotic-chemoattractant treatment alone exhibited a combination of reduced bacterial load and improved tissue pathology.

## Discussion

In this work, we successfully explored the concept of antibiotic-chemoattractants to enhance the neutrophil response against the deadly immune-evading pathogen *S. aureus*. In choosing vancomycin as the targeting moiety in these constructs, we were able to take advantage of its cell wall binding properties to establish a gradient. Furthermore, the clinical use and development of glycopeptide antibiotics such as vancomycin provides avenues for improving the properties of these agents and also offers the potential for scaled production in the future[19]. As a chemotactic agent, the use of fPeps provided the opportunity to explore a wide compound space in a facile manner using the solid-phase synthesis of peptide libraries, which showed that modified sequences display different kinetics of neutrophil recruitment and phagocytosis for different antibiotic-chemoattractants. This offers the possibility to tune these responses as a function of the fPep payload sequence, with the most promising conjugates fMChaFG (**C11**) not only stimulating effective chemotaxis but also increasing phagocytosis per neutrophil for all donors.

FPR are promiscuous receptors that bind to a range of ligands that are structurally and functionally distinct. Biased agonism or ligand-dependent agonism is an emerging field of GPCR research, which has the potential for the development of drugs with selective beneficial effects. We have previously demonstrated that

biased agonism is an important molecular pharmacology consideration for these receptors[49,52], with biased agonists for FPRs known[49]. This provides the potential for designer FPR ligands to bias aspects of FPR activation. Understanding how altered fPep sequences lead to differential FPR binding and intracellular signaling is highly important in understanding their effects on the host defense, potentially as means to decode "danger sensing". Activation of these FPRs leads to not just neutrophil chemotaxis but also to release of various compound (reactive oxygen species, host defense peptides, chemokines), formation of neutrophil extracellular traps[53] and enhance phagocytosis of *S. aureus*[14,41]. How FPRs (in particular FPR2) transmit signals from their diverse ligands to the intracellular space to play distinct, sometimes opposite, physiological roles in inflammation is largely unknown. Here, we observed that whilst the widely investigated fPep sequence fMLFG led to FPR1 activation as anticipated, substitution of the second Leu residue with Cha (fMChaFG) was sufficient to alter the activation profile of this peptide to favor FPR2 activation. However, upon conjugation to vancomycin, the fMChaFG conjugate was unable to compete effectively with the known FPR2 agonist, RhB-PB10. These results suggest a complex relationship between receptor binding and downstream activation events, which require further investigation to determine how the fMChaFG conjugate affords enhanced phagocytosis. This process will involve untangling the tangled web of ligand recognition, biased signaling, allosteric modulation, receptor cross-talk, as well as hetero and homo receptor dimerization in these important G-coupled protein receptors[54].

We also noted differences between the human donors in terms of neutrophil chemotaxis and phagocytic responses to the different fPep sequences (in particular fMLFL and fMLYG). These different donor responses could be explained by the variation in FPR number and function that are present in the human population due to both polymorphisms within the *FPR1* gene and the three isoforms of FPR1[46]. These genetic variations lead to differences in response to fPep sequences[55,56] and in turn the FPR encoded has therefore been linked to the response of individuals to infections[57–59]. The modular nature of the antibiotic-chemoattractants provides a potential pathway toward personalized treatment of an infection that takes into account the FPR genotype to select the optimum fPep sequence for the patient. This complex chain of effects also highlight the importance of using the correct tools to validate the multifaceted activities of these immunotherapeutics, which are otherwise lost in approaches that utilize "averaged" donor responses.

The infection-on-a-chip data allowed us to identify the optimal antibiotic-chemoattractant to test in a mouse model of MRSA pneumonia. This revealed that antibiotic-chemoattractants can function in the whole host environment, and at a five-fold lower concentration compared to standard vancomycin treatment for

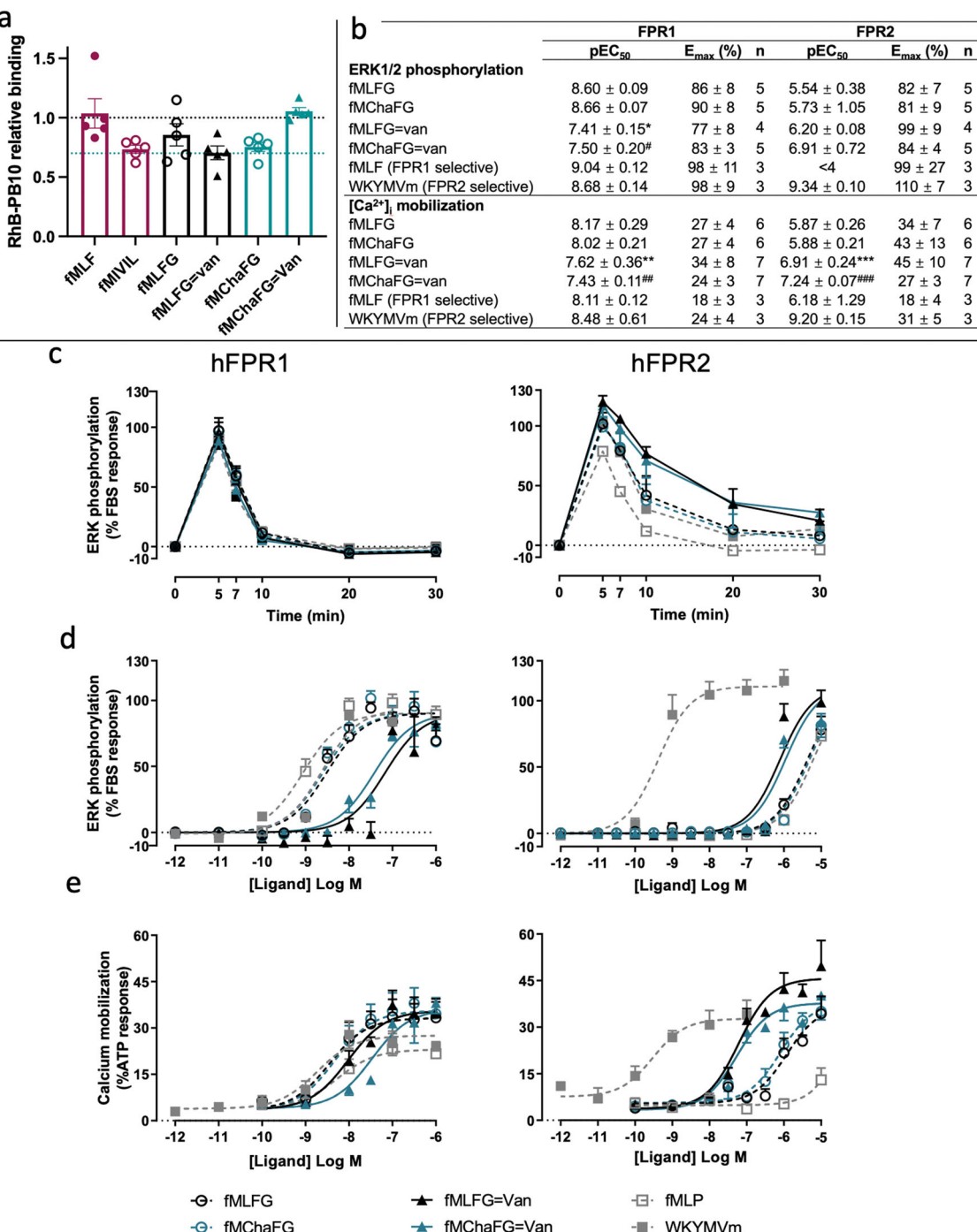

**Fig. 4 Interaction of fPeps with human FPRs and activation of downstream pathways. a** Competition binding to human neutrophils between the FPR2 antagonist RhB-PB10 and different fPeps. fMLF binds preferentially to FPR1 and was used as a negative control, while fMIVIL binds preferentially to FPR2. The peptides fMLFG and fMChaFG and their corresponding conjugates were examined for their ability to compete with RhB-PB10 binding to neutrophils, $n = 5$ biologically independent experiments, with error bars of SEM. **b** CHO cells overexpressing human FPR1 or 2 were assessed for the downstream activation of either ERK1/2 phosphorylation or Ca$^{2+}$ mobilization ([Ca$^{2+}$]$_i$)with E$_{max}$ and pEC$_{50}$ calculated. Data presented as mean ± SEM and an unpaired $t$ test were used to compare the effects of the fMLFG and fMChaFG in free and conjugated forms. n indicates the number of biological replicates with two-tailed p-values indicated by *$p = 0.0002$, **$p = 0.02$, ***$p = 0.01$ vs. fMLFG, #$p = 0.0006$, ##$p = 0.01$, ###$p = 0.000048$ vs fMChaFG. **c** Time course of ERK1/2 phosphorylation ($n = 4$ biological replicates performed in triplicate, error bars are SEM). **d** Concentration-response curves for pERK1/2 were determined at 5 min, the peak-response time point determine during the time course. The positive controls were the hexapeptide Trp-Lys-Tyr-Met-Val-D-Met (WKYMVm), a specific FPR2 agonist, along with N-formyl-Met-Leu-Phe (fMLF) as a specific FPR1 agonist. Data were normalized to vehicle and the response elicited by 10% FBS, with nonlinear regression curve plotted with error bars of SEM, ($n = 3-5$ experiments, each performed in duplicate). **e** Intracellular calcium mobilization was determined in response to fPeps in CHO cells overexpressing either human FPR1 or FPR2. The fluorescence signal was normalized to the cellular response mediated by the positive control, adenosine triphosphate (ATP) at 100 μM, plotted with nonlinear regression curve, with error bars of SEM, ($n = 3-7$ experiments, each performed in triplicate).

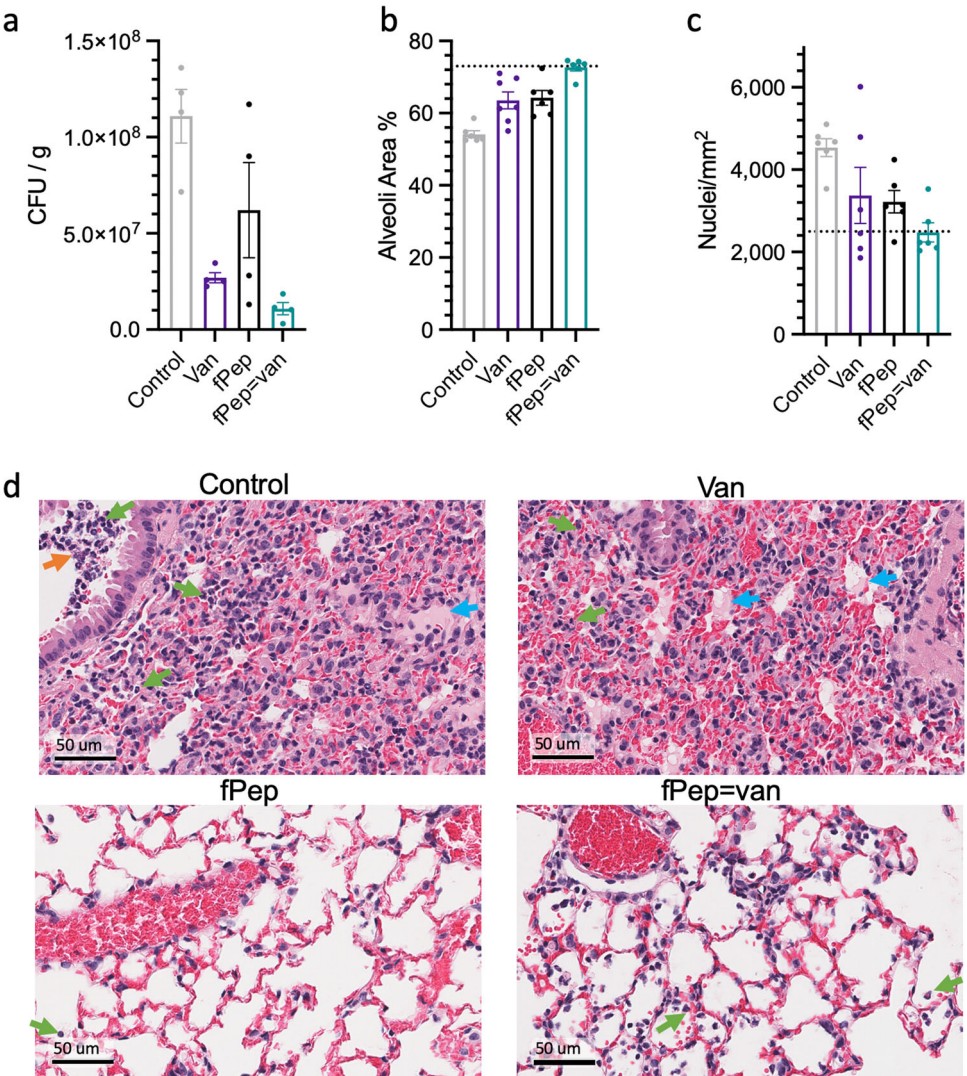

**Fig. 5 fPep-vancomycin conjugate C11 reduces bacterial load and inflammation in a mouse pneumonia model.** Eight-week-old female mice were infected by intranasal inhalation of $10^7$ CFU *S. aureus* to induce pneumonia. One-hour postinfection (hpi) mice were given intranasal therapy at 0.2 mg/mouse equivalent of vancomycin or vehicle control (control). The lungs were collected at 12 hpi, and the bacterial load in the lung tissue of four mice per treatment group was determined (**a**). The fMChaFG conjugate (**C11**, fPep = van) resulted in half the bacterial load compared to vancomycin (van) alone ($n = 4$ mice/group, error bars are SEM). The percentage of alveoli space (**b**) or the number of nuclei (**c**) was determined for hematoxylin and eosin-stained histology lung samples (**d**) from 6 fields of view of the outer lung lobes. Uninfected mouse lung is indicated by dotted line, error bars are SEM. Green arrows indicate neutrophils, blue arrows proteinaceous fluid, and orange arrows indicate cocci bacteria.

the clearance of MRSA pneumonia infections. This efficacy likely stems from the presence of a complete cohort of immune cells expressing FPR in the whole animal, which would contribute to the twofold enhanced clearance of the infection we observed in mice lungs. While we have focused on neutrophils in this study, FPR-expressing cells also include other important innate immune cells - macrophages, monocytes, and dendritic cells[46], along with vascular cells[60]. It is important to note that in mice, fPep sequence specificity is different despite mice and human sharing high sequence homology between FPRs[61]. However, in combination with the infection-on-a-chip approach, these data provide strong support for the efficacy of our antibiotic-chemoattractants as a potential new therapy to overcome resistant bacterial infections.

Taken together, the fusion of an antibiotic targeting element with immune stimulation can provide a viable immunotherapeutic strategy to treat resistant *S. aureus* infections. Given the recognized problems with our reliance on natural antibiotics for

infection control, it is vital that we explore new strategies to be able to control these resistant and highly virulent bacterial strains that threaten the cornerstones of modern medicine. Thus, the development of antibiotic-chemoattractants is not only important as an example of a potential strategy to control *S. aureus* infections but also provides a template for the experimental pipeline needed to identify and investigate these complex agents. The future of this approach lies in understanding the differences seen in host responses to these compounds and further exploring the ability of broadly effective compounds—such as the conjugates containing fMChaFG—to resolve different types of infections. Additionally, unlocking the specificity that results in not just chemotaxis, but also enhances neutrophil killing will be vital to ensure the success of such immunotherapeutics. The antibiotic-chemoattractants developed in this work demonstrate the potential to rejuvenate current antibiotics, and also to deliver a more nuanced approach to infection control by application of a suite of tools to assess the complex activity of these compounds.

## Methods

**Vancomycin functionalization.** The C-terminal carboxylic acid of vancomycin HCl (1 eq) was activated with COMU (1 eq) and triethylamine (TEA, 1 eq) in dimethylformamide (DMF) at ~30 mM[27]. After 5 min of pre-activation, either 3-azidopronamine (2 eq) or propargylamine (2 eq) were added. The reaction was stirred overnight, analyzed by LCMS (see below for conditions) after which the product was precipitated in diethyl ether and washed twice with diethyl ether, giving crude product in ~80% yield that was purified by preparative RP-HPLC (see below for conditions). 5-azidopentanoic acid (1.1 eq) was pre-activated with COMU (1.1 eq) and TEA (2 eq) in DMF for 5 min. This was added to a solution of vancomycin (HCL salt, 1 eq) dissolved in DMF at ~20–50 mM, stirred overnight and subsequently analyzed by LCMS. The product was precipitated in diethyl ether (Et$_2$O) and washed twice with Et$_2$O, giving crude product in ~85–98% yield. LC analysis allowed us to determine that the ratio between four compounds (Supplementary Fig. 5), namely vancomycin, azidopentanoyl loaded on the N-terminal, azidopentanoyl loaded on the vancosamine moiety and the compound resulting from double addition (30%, 40%, 10%, 20%) that were all isolated by preparative RP-HPLC. N-terminal and vancosamine conjugation reactions were performed in one pot without protection; the resultant regioisomers were separated using RP-HPLC, with the identity of each compound confirmed by small-scale hydrolysis. The resultant mass difference caused by the loss of either vancosamine or the modified vancosamine thus allowed the identity of each compound to be confirmed.

**Solid-phase peptide synthesis on SynPhase™ lanterns and rink-Amide resins.** Peptides were synthesized on SynPhase™ PSDRAM lanterns and Rink-amide resin using an Fmoc strategy[62]. Deprotection steps were carried out in 20% piperidine/DMF for 20 min, followed by four DMF washes (3 min for lanterns and 30 s for resins). Coupling was carried out with 4 eq. amino acid, 3.8 eq. HCTU, and 8 eq. DIEA for 1 h with shaking, followed by four DMF washes as described above. The N-terminal formylation was carried out after the final deprotection step by coupling formic acid (3 eq), with pre-activation achieved in the presence of EDC.HCl (3 eq) and HOBt (3 eq) in DCM for 15 min at 4 °C. Afterwards, DIEA (1 eq) was added to maintain a basic pH for the coupling to proceed and the reaction left overnight at room temperature. For the formylated peptide library, a split/mix method was performed where lanterns were mixed for common steps and split for the coupling of specific amino acids as required. This was made possible due to color tagging of the lanterns. Each peptide was separately cleaved using TFA/H$_2$O/TIS (95/ 2.5/ 2.5; v/ v'/ v") for 1 h, concentrated under N$_2$, precipitated in cold Et$_2$O, washed three times with cold Et$_2$O and air-dried overnight. The isolated peptides were analyzed by LCMS and purified by preparative RP-HPLC, leading to the isolation of product with >95% purity as assessed by RP-HPLC with yield of 5–20% (lanterns) and 25–45% (resins) across all steps.

**Formylation, formylated-methionine coupling and acetylation on solid support.** Formylation and formylated-methionine coupling were carried out after the final deprotection step as follows: Formic acid or fMet-OH (3 eq) and HOSu (3 eq) were dissolved in DCM at 0 °C with stirring. EDC.HCl (3 eq) was added gradually to the solution before the lanterns were added. DIEA (1 eq) was then added to maintain a basic pH; coupling was allowed to proceed overnight at room temperature. Acetylation was carried out in Ac$_2$O/DIEA/DMF (5:5:90) with shaking overnight following the final deprotection step.

**Vancomycin-peptide conjugation.** Reactions were performed in 1.5 mL tubes at ~3 mM concentration. The formylated peptide (1 eq, R1–13, Supplementary Fig. 6) was dissolved in DMF (50 μL/mg) and added to azide-functionalized vancomycin (1.1 eq., GPA1–4, Supplementary Fig. 5) dissolved in aqueous sodium ascorbate buffer (22 mM, pH 8.0)[27]. Next, an aqueous solution of CuSO$_4$ (3 eq, 0.1 M) was added, leading to the rapid formation of precipitate. The reaction was monitored by LCMS after shaking for 1 h and the product purified directly by preparative RP-HPLC, affording >95% purity, and overall yield of in 25–47% (Supplementary Fig. 7).

**LC/MS analysis conditions.** Analyses were carried out on a Shimadzu High-Performance Liquid Chromatograph coupled to a Mass Spectrometer LCMS-2020 (ESI, operating both in positive and negative mode) equipped with an SPD-20A Prominence Photo Diode Array Detector and a LC-20AD solvent delivery module running lab solutions v5.97SP1. Analytical separations were performed using a Waters XBridge BEH300 Prep C18 column (10 μm, 4.6 × 250 mm). The solvents used were water + 0.1% formic acid (solvent A) and HPLC-grade ACN + 0.1% formic acid (solvent B).

**Preparative RP-HPLC conditions.** Purification using RP-HPLC was performed using a Shimadzu High-Performance Liquid Chromatograph equipped with a SPD-M20A Prominence Photo Diode Array Detector and two LC-20AP pumps running lab solutions v5.97SP1. Preparative separations were performed on a Waters XBridge BEH300 Prep C18 column (5 μm, 19 × 150 mm) at a flow rate of 10 mL/min. The solvents used were water + 0.1% formic acid (solvent A) and HPLC-grade ACN + 0.1% formic acid (solvent B).

**High-resolution mass spectrophotometry analysis of conjugates.** High-resolution mass spectrometry measurements were performed on an Orbitrap Fusion mass spectrometer (Thermo Scientific) coupled to a Dionex UltiMate 3000 RSLCnano system equipped with a Dionex UltiMate 3000 RS autosampler, an Acclaim PepMap RSLC analytical column (75 μm x 50 cm, nanoViper, C18, 2 μm, 100 Å; Thermo Scientific) and an Acclaim PepMap 100 trap column (100 μm x 2 cm, nanoViper, C18, 5 μm, 100 Å; Thermo Scientific). The peptides were separated by increasing concentrations of 80% acetonitrile / 0.1% formic acid at a flow of 250 nl/min over 30 min. The instrument was operated in alternating data-dependent acquisition (DDA) and parallel reaction monitoring (PRM) cycles in such that five ms2 scans were triggered per survey ms1 scan followed by several targeted ms2 scans to ensure fragmentation of predefined, sample-dependent m/z precursors. Each survey ms1 scan (300–1800 m/z) was acquired with a resolution of 240,000 and a normalised AGC (automatic gain control) target of 200%. Dynamic exclusion was set to 10 s after one occurrence. The five most intense ions were selected for HCD fragmentation (fixed collision energy mode, 24% HCD Collision Energy) with a resolution of 15,000, a normalised AGC target of 200% and a fixed first mass of 100 m/z. Subsequent targeted ms2 scans were acquired with essentially identical settings. The raw data files were analysed with Qual-Browser (XCalibur 3.0.63, Thermo Scientific) to view spectra and to generate extracted ion chromatograms. The HR-MS data have been deposited to the ProteomeXchange Consortium via the PRIDE partner repository. Data are available via ProteomeXchange with identifier PXD028332.

**In silico calculations performed for the fPep library.** All formylated peptides were represented in Chemdraw (v19), which was used to calculate the logP for each peptide. Additionally, the SMILES codes were used in Elbow-Phoenix (v1.19.2) to perform a simple minimisation and generate their respective.pdb files. Afterwards, the minimised fPeps.pdb were opened in Chimera (v1.15) that allowed the calculation of the total solvent-accessible surface area for each fPeps (Tools/ Surface/ binding analysis/ Measure volume and area).

**Bacterial isolates and media.** All bacterial isolates used in this study are listed in Supplementary Table 1. The American Type Culture Collection strain (ATCC 29213) and *S. aureus* strains were cultured onto brain heart infusion (BHI) agar (BD) or into broth (BD) and grown at 37 °C with shaking. *S. aureus* was grown in cation adjusted Muller Hinton Broth II (CAMHB, Becton Dickson) for testing. All bacterial strains were stored in glycerol broth at −80 °C.

**Imaging vancomycin binding to *S. aureus* strains.** The imaging study was designed to compare the binding interaction to different bacteria by fPep, vancomycin and a fPep-vancomycin conjugate. To minimize the effect of the bulk of the fluorescent dye interfering with vancomycin binding to the bacteria, the fluorescent dye BODIPY was introduced at the C-terminal of vancomycin and fPep-vancomycin conjugate. BODIPY® FL NHS Ester (1.1 eq.) dissolved in DMF (5 mM) was added to R11 leading to B1 or propargylamine leading to a BODIPY-alkyne compound; the addition of TEA (1 eq.) allowed the completion of the coupling reactions. The labelled vancomycin (B2) and fPep-vancomycin conjugate (B3) were generated using click chemistry between the BODIPY-alkyne and GPA2 for B2 and B1 and GPA2 for B3 (Supplementary Fig. 9).

*S. aureus* strains (MSSA, ATCC 29213; MRSA, A8090) along with *B. subtilis* and *E. coli* (DH5 alpha) were grown to mid-exponential phase and incubated with 16 μM of the BODIPY-labelled vancomycin (B2); the BODIPY-labelled fPep (B1); or the antibiotic-chemoattractant labelled with BODIPY (B3) for 30 min at room temperature. The cells were collected by centrifugation (370 *g*) and washed three times with media to remove the excess unincorporated compound, followed by resuspension in fresh MHB broth. The cells were then imaged using a Zeiss LSM 980 Airyscan 2 (Carl-Zeiss, Jena, Germany) with a 63x oil C-PlanApo 1.4NA objective using the optimal super-resolution mode. Images were processed in FIJI v2[36] with intensity set across all images. At least three biological replicates were imaged.

**Image analysis of BODIPY-labelled vancomycin binding to *S. aureus* strains.** *S. aureus* strains A8090 (MRSA), and ATCC 29213 (MSSA) incubated with either fluorescently labeled vancomycin or vancomycin conjugated to fMLFK were analyzed using the FIJI image processing package within ImageJ[1]. Plots of the fluorescence intensity for each cell across the septum and cell walls were determined in triplicate for each cell and the raw data was recorded; more than 120 cells were analyzed per treatment across three biological replicates. The ratios of peak fluorescence values for each cell (wall and septum) were calculated and then aggregated to find the average ratio value for each cell. The aggregated data was visualized using the Graphpad Prism (v9).

**Antimicrobial assay.** Antimicrobial activity of the antibiotic-chemoattractants was determined as per the Clinical and Laboratory Standards Institute guidelines for microbroth dilution assays. Compounds stocks were prepared as two-fold serial dilution at 10x stock concentration required (final concentration of DMSO 0.5%). Growth of the *S. aureus* ($5 \times 10^5$ CFU/mL in CAMHB) was monitored in 96-well flat bottom polypropylene microtiter plates (Greiner) covered with BreatheEasy

membranes (Diversified Biotech) and absorbance of each well was measured at 595 nm every 4 h for 24 h (Clariostar plate reader, software v5.20R5, BMG). The percentage growth was calculated relative to the no compound control and the average of three biological replicates plotted using GraphPad Prism 9 software. The minimum inhibitory concentration was determined by the lowest drug concentration that inhibited bacterial growth to 90% compared to the 0 μM control[63].

**Neutrophil isolation.** For neutrophils used in transwell assays whole blood was collected in the presence of EDTA, under approval from Monash University Human Research Ethics Committee (project #9572) with written informed consent from healthy human donors; donors were de-identified.

For neutrophils used in microfluidic devices, de-identified fresh blood samples (10 mL) were purchased from Research Blood Components where blood samples were obtained from healthy volunteers aged ≥18 years who were not receiving immunosuppressant agents. Additionally, venous blood samples from healthy volunteers were collected by phlebotomy, after receipt of written informed consent, and under approval by the MGH Institutional Review Board (protocol 2008-P-002123). Neutrophils were isolated from whole blood using EasySep direct human neutrophil isolation kit (StemCell Technologies) as per the manufacturer's instructions.

**Transwell chemotaxis assay.** The chemotaxis assay was performed in a 96-transwell plate with 3 μm pores (Corning). The plate was prepared by adding 200 μL of peptide in chemotaxis buffer (4 mM L-glutamine, 0.5% Human Serum Albumin, Sigma, 49% Roswell Park Memorial Institute (RPMI) 1640, 49% Hank's buffered salt solution, ThermoFisher) into the bottom receiving wells at a final peptide concentration of 1, 10, 100, and 1000 nM (0.02% DMSO). The negative control of 0.02% DMSO was used. The top of the transwell plate was loaded with 200,000 neutrophils in 75 μL chemotaxis buffer. The plate was incubated in the dark at 37 °C with 5% $CO_2$ for 1.5 h. Neutrophils that had migrated into the bottom receiver plate were lysed by adding 10 μL/well of elastase assay buffer (500 mM Tris-HCl, 1 M NaCl, pH 7.4, and 0.5% (v/v) Triton X-100), releasing neutrophil elastase. Elastase activity was used as an indicator of neutrophil numbers and was measured by adding the chromogenic elastase substrate N-methoxysuccinyl-Ala-Ala-Pro-Val-p-nitroanilide (Sigma) at a final concentration of 1 mM[35]. After 30 min at room temperature, the absorbance at 405 nm was determined using a microplate reader (Clariostar plate reader, BMG). All peptides were tested in duplicate with 3–4 biological replicates, ensuring different donors each time.

Human neutrophils displayed a variability of response to formylated peptides between individual donors. This variability was taken into account by determining the percentage of neutrophil recruitment relative to the recruitment observed to the fMLFG peptide (**FP1**) at 100 nM from the same neutrophil donor. To compare data between donors, the positive control of 100 nM fMLFG (**FP1**) was set as 100% and all other conditions calculated relative to this. In addition, testing of the entire fPep library or the conjugated library was performed at the same time to allow for comparisons to be made across the results from 3 neutrophil donors. Data were plotted using GraphPad Prism 8 software.

**Preparation of microfluidic devices.** Microfluidic devices utilized were previously described by Ellett et al.[45]. Briefly, silicon wafers were prepared by sequentially spin-coating two layers with negative photoresist (SU-8, Microchem, Newton, MA), patterning by exposure to UV through two chrome lithography masks (Front Range Photo Mask, Palmer Lake, Colorado), and processing according to manufacturer instructions. The first layer, making up the migration channels, was 5 μm thick, while the second layer, making up the outer and inner chambers, was 50 μm thick. PDMS (polydimethylsiloxane, Fischer Scientific, Fair Lawn, NJ) devices were fabricated through soft lithography techniques, using the pattered wafer as a negative mold. PDMS base elastomer was mixed thoroughly with curing agent at a ratio of 10:1, poured onto the wafer, and degassed for at least 1 h in a vacuum chamber. After baking overnight at 75 °C, the cured PDMS was cut from the silicon wafer with a scalpel and the inlets and outlets punched using a 1.0 mm biopsy punch (Harris Uni-Core™). Individual devices were then cut free using a scalpel, treated with oxygen plasma, and bonded irreversibly to glass-bottom well plates (MatTek, Ashland, MA) by heating to 75 °C for 10 min.

Use of this microfluidic device for monitoring migration of neutrophils and control of the *S. aureus* growth in cell culture medium has been outlined in detail previously[45]. Briefly, microfluidic devices were primed with *S. aureus* (SH1000-GFP, $1 \times 10^6$/mL) in culture media (Iscove's Modified Dulbecco's Medium + 10% Fetal Bovine Serum) by loading the outer chamber and applying vacuum to draw bacteria into the inner chambers. The outer chamber was then washed by pushing 200 μL of media through the device using a pipette, and loaded with neutrophils ($5.5 \times 10^6$/mL, stained with Hoechst 33342 dye, 10 min at 32 μM). Additionally, neutrophil phagocytosis was monitored by using pHrodo Red *S. aureus* BioParticles® (Life Technologies) in replace of live *S. aureus* cells. BioParticles were used at a final concentration of $1 \times 10^7$/mL, which gave 50–100 BioParticles within the microchamber. The concentration of cells were determined by hemocytometer.

**Microfluidic data acquisition and analysis.** Kinetics of neutrophil migration, *S. aureus* growth and phagocytosis was determined using imaging with an automated Nikon Eclipse TiE inverted wide-field microscope with stage control and perfect focus system run through the Nikon NIS element v4. The biochamber was maintained temperature at 37 °C and $CO_2$ at 5% for the 8 h of the experiment, where images were captured every 5 min for each field of view (FOV), with at least 4 FOV captured per experimental condition. For each experimental condition at least 12 chambers were tracked for neutrophil migration using the FIJI[64] plugin Trackmate[65]. The percentage of neutrophils recruited into the chamber was determined relative to the total number of neutrophils in each FOV divided by the number of chambers visible. Data were averaged and the results from at least 3 donors plotted. The pHrodo or *S. aureus* GFP fluorescence within the microchamber was calculated by thresholding the image and measuring the area in FIJI[36]. Data were plotted using GraphPad Prism 8 software (GraphPad Software, San Diego, CA, USA).

**FPR competition binding assay.** A competition binding assay was performed based modification of the previous method[66]. To determine the interaction of the formylated peptides with FPR2 of human neutrophils the fluorescently labelled FPR2 antagonist RhB-PB10 was used. RhB-PB10 was synthesized by solid-phase peptide synthesis on Wang resin. After initial acylation with Fmoc-L-Arg(Pbf)-OH, peptide elongation was performed using a solution of 20% piperidine for Fmoc removal and DIC/ Oxyma for Fmoc-amino acid or Rhodamine B coupling. After completion of the sequence, the peptide was cleaved from the resin and protecting groups were removed via TFA cleavage. Following concentration, the peptide was precipitated using diethyl ether, collected by centrifugation on a spark-resistant centrifuge and purified using RP-HPLC.

RhB-PB10 (15 μM) was preincubated with human neutrophils ($5 \times 10^6$ neutrophils/mL) in Bis-Tris buffer pH 6 for 1 h. Neutrophils were washed twice in Bis-Tris buffer pH 6 to remove any unbound RhB-PB10, before being incubating for 5 min with 8 μM of test peptide (fMLFG and fMChaFG both free [**FP1** & **FP11**] and conjugated [**C1** & **C11**]) or the control peptides fMLF (Auspep), a specific FPR1 agonist, or fMIVIL (Auspep), an FPR2 specific agonist. Any RhB-PB10 that was competed off by the peptides was then removed by washing in Bis-Tris buffer. Rhodamine B fluorescence levels of each sample were determined using microplate reader (Clariostar plate reader, BMG) with ex 550 nm, em 585 nm.

**Measurement of ERK phosphorylation in response to FPR1 or FPR2 activation.** FlpIn Chinese hamster ovary (CHO) cells stably expressing either the human FPR1 or FPR2. Both of the cell lines were cultured in Dulbecco's modified eagle medium (DMEM), which contained 10% foetal bovine serum (FBS, JRH Biosciences) and 500 μg/mL of the antibiotic hygromycin-B (Invitrogen Inc). Cells were grown at 37 °C in a humidified incubator containing 5% $CO_2$ as described previously[67]. Cells were seeded into 96-well tissue culture plates ($4 \times 10^4$ cells/well) and after 6–8 h, cells were washed with PBS twice (50 μL/well in each washing step) and then starved by incubating in 90 μL/well of serum-free DMEM overnight at 37 °C in 5% $CO_2$ before assaying for ERK1/2 phosphorylation (pERK1/2) and intracellular $Ca^{2+}$ mobilization.

A time course was used to assess peak pERK1/2 phosphorylation for all the compounds. The positive control of 10% FBS, and serum-free DMEM as vehicle control. FPR specific positive controls were used of FPR1 agonist N-Formyl-Met-Leu-Phe (fMLF, Sigma-Aldrich) and the FPR2 agonist, the hexapeptide Trp-Lys-Tyr-Met-Val-DMet (WKYMVm, Tocris Bioscience). All the compounds were tested with time points taken at 0, 5, 7, 10, 20, 30. The time versus concentration-response curves were generated and the peak-response was observed at 5 min for all agonists.

For both the time course and concentration-response curves, the stimulation was terminated by the rapid removal of serum-free DMEM with agonists and adding 100 μL/well of SureFire lysis buffer. The detection step was performed according to the manufacturer's instructions (AlphaScreen detection kit, PerkinElmer Life and Analytical Science)[49]. Data were normalized to the vehicle (serum-free DMEM) and the response elicited by 10% FBS, with nonlinear regression performed using GraphPad Prism 9.0.

**Measurement of calcium mobilization in response to FPR1 or FPR2 activation.** For the calcium mobilization assay, the cells were plated and starved in the same manner as the pERK1/2 assay, before washing the cells once in HEPES buffered saline (HBS) solution [150 mM NaCl, 1.3 mM CaCl$_2$, 5 mM KCl, 1 mM MgSO$_4$.7H$_2$O, 1.5 mM NaHCO$_3$, 10 mM HEPES, and 10 mM D-glucose; pH 7.4]. Then the cells were incubated at 37 °C under 5% $CO_2$ condition with HBS containing 0.5% (w/v) bovine serum albumin (BSA), 4 mM probenecid and 1 μM Fluo-8-AM (Abcam) for 1 h. FlexStation three plate reader (Molecular Devices) were used to determine $Ca^{2+}$ mobilization. The peak change in fluorescence signal was normalized to the vehicle (HBS solution), and the cellular response mediated by the positive control, adenosine triphosphate (ATP, 100 μM), and fitted with nonlinear regression curve in GraphPad Prism 9.0.

**Quantification of FPR agonist potency and maximal agonist response.** Nonlinear regression was performed using GraphPrism 9.0. Concentration-response

curves mediated by each agonist across the two signaling pathways (intracellular $Ca^{2+}$ mobilization, pERK1/2 phosphorylation), were fitted to derive the maximal agonist effect ($E_{max}$), normalized to the indicated positive control, and the ligand potency ($pEC_{50}$), defined as the negative logarithm of the agonist concentration that gives a response halfway between the lower and upper asymptotes of the concentration-response. Significance was determined using Student's unpaired $t$ test, with two-tailed $p$ values of $<0.05$ considered statistically significant.

**Mouse pneumonia model.** All experiments were performed in accordance with the Animal Research Ethics Committee at Monash University (MARP/2018/012). Mice were housed with 12/12 h light-dark cycle with room temperature between 18 and 24 °C, and 40 and 70% humidity. Eight-week-old female Balb/c mice were anaesthetized with 4% isoflurane, before intranasal inhalation of $10^7$ CFU *S. aureus*, in 50 µL of PBS[68]. The concentration of bacteria was important to establish this model in non-neutropenic mice, as too little of an inoculum was easily cleared. In addition, too high of an inoculum with this highly virulent strain resulted in a severe infection reaching protocol determined endpoints within 12 h. At 1 hpi mice were treated intranasally with 50 µL of vancomycin (0.2 mg/50 µL) or equivalent molar amount of fMLFG (**FP1**), fMChaFG conjugate (**C11**) or vehicle control (5% DMSO in PBS). Mice were humanely euthanized by $CO_2$ inhalation at 12 hpi and lung bacterial density was assessed from four mice per treatment group. Lungs were homogenized and underwent serial dilution and plating onto solid media for CFU enumeration. Significance was determined using a one-way ANOVA with $p < 0.05$ being significant and calculated using GraphPad Prism 8 software.

For pathology assessment, excised lungs from one mouse for each of the treatment group was fixed in 10% buffered formalin before undergoing routine histological processing (Monash University Histology Platform). Lungs were embedded in paraffin, cut into 5 µm thick sections and stained with hematoxylin and eosin. Tissue pathology was assessed by a veterinary pathologist (Dr Mark Williamson, Gribbles Veterinary Pathology, Clayton, Australia).

The loss of alveoli structure was quantified in the H&E sections by determining the area of white space in six FOV ($1.8 \times 1.2$ mm) in the outer lobes of the lung (threshold of 0–240). In addition, the number of nuclei in these FOV were also determined. This was performed by isolating nuclei in the image (threshold of 0–130) and then using the analyze particle function (particles 80–4000, circularity 0.05–1) in FIJI[36] to count the number of nuclei detected. Data were plotted using GraphPad Prism 8 software.

**Reporting Summary**. Further information on research design is available in the Nature Research Reporting Summary linked to this article.

## Data availability

Source data for the figures are provided in the Source Data file. Microscope imaging can be provided on reasonable request to the authors. The high-resolution mass spectrometry data for the compounds have been deposited to the ProteomeXchange Consortium via the PRIDE[69] partner repository with the dataset identifier PXD028332. Source data are provided with this paper.

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

## Acknowledgements

The authors acknowledge the facilities and scientific and technical assistance of the Monash Histology Platform, Department of Anatomy and Developmental Biology, Monash University. Tissue pathology was assessed by a veterinary pathologist Mark Williamson, Gribbles Veterinary Pathology, Clayton, Australia. The authors acknowledge the scientific and technical assistance of the Monash Micro Imaging facility, along with the Monash Animal Research Platform at Monash University, Victoria, Australia. This study used BPA-enabled (Bioplatforms Australia) / NCRIS-enabled (National Collaborative Research Infrastructure Strategy) infrastructure located at the Monash Proteomics and Metabolomics Facility.

Microfabrication of microfluidic devices was conducted at the BioMEMS Resource Center at Massachusetts General Hospital, and supported by a grant from the National Institutes of Health (Grants GM092804 and EB002503) and Shriners Hospital for Children. FE was supported by fellowships from the American Australian Association, Shriners Hospital for Children, and the Executive Committee for Research at the Massachusetts General Hospital.

JAEP was supported by the 2017 Victoria Fellowship from Victorian Endowment for Science, Knowledge and Innovation (VESKI, Australia) and a Hugh Rodgers fellowship from the Melbourne Boston Sister City Association to conduct this research in Boston at Massachusetts General Hospital and Harvard Medical School. MJC is supported by an Australian National Health and Medical Research Council Career Development Fellowship (APP1140619). MJC and JAEP work was supported by Ideas grant from Australian National Health and Medical Research Council (APP2003325). MJC, JAEP and JT are supported by the Australian Research Council Centre of Excellence in Peptide and Protein Science (CE200100012).

AYP is supported by an Australian National Health and Medical Research Council Practitioner Fellowship (APP1117940). GJL was supported by an Australian National Health and Medical Research Council Senior Research Fellowship (APP1044754). The Australian Regenerative Medicine Institute is supported by grants from the State Government of Victoria and the Australian Government. TF supported by a Faculty of Pharmacy and Pharmaceutical Sciences (FPPS) Graduate Research Scholarship and a FPPS International Graduate Research Scholarship provided by Faculty of Pharmacy and Pharmaceutical Sciences, Monash University. While CXQ received a National Heart Foundation Future Leader Fellowship and an Australian National Health and Medical Research Council Grant (APP1187989).

## Author contributions

J.A.E.P. designed and performed experiments for the testing of compounds in neutrophil recruitment, antimicrobial, and in vivo assays. Performed data analysis and wrote the manuscript. J.T. designed and synthesis of the antibiotic-chemoattractants conjugates and fluorescent compounds, along with in silico analysis of fPep library, edited manuscript. F.E. designed and fabricated the microfluidic devices along with designing microfluidic experiments, edited manuscript. X.K. designed and performed the mouse pneumonia model, edited manuscript. A.J.F. performed AiryScan microscopy and data analysis, edited manuscript. T.F. performed F.P.R. activation assays R.L. constructed and tested the fPep library, edited manuscript. A.T. performed image analysis, edited manuscript. S.L. performed microfluidic image analysis and transwell assays, edited manuscript. S.A.W. synthesized BODIPY-labelled compounds, edited the manuscript. R.B.S. performed and analysed HRMS, edited manuscript. G.L. conceptualized experiments to test neutrophil migration and phagocytosis, edited manuscript. C.X.Q. conceptualized F.P.R. experiments and edited manuscript. D.I. designed microfluidic experiments, edited manuscript. A.Y.P. conceptualized experiments and compound design, edited manuscript. M.J.C. developed the concept for the work and assisted in experimental and compound design; writing of the manuscript.

## Competing interests

A.Y.P., J.A.E.P., J.T., and M.J.C. have filed for patent protection (PCT/AU2021/050306) of the antibiotic-chemoattractant compounds. The remaining authors declare no competing interests.
