## [Peer Review File · Nature Communications]

Antibiotic-Chemoattractants enhance neutrophil clearance of *Staphylococcus aureus*REVIEWER COMMENTS

Reviewer #1 (Remarks to the Author):

In this manuscript, the authors describe the synthesis and application of formyl peptide covalently linked to vancomycin to operate as a dual function antibiotic-chemokine agent. The vancomycin component was intended to provide a higher local concentration of the formyl peptide, which upon activation of FPRs will result in the chemotaxis of immune cells (e.g., neutrophils). This idea appears to be fairly novel against Gram-positive pathogens and while it would likely lose its efficacy against vancomycin-resistant organisms, it should still work well with drug-sensitive pathogens. Similar concepts (antibiotic-formyl peptide conjugates) were developed against Gram-negative but were not mentioned (*Antimicrob Agents Chemother.* 2005 Aug; 49(8): 3122-8. doi: 10.1128). The authors first tested three points of modification as noted by V, C, and N and these correspond to three well-established points of tolerated modifications on vancomycin with a fluorophore and proceeded to evaluate chemotaxis with various formyl peptide sequences. Ultimately, they settled on a few sequences that demonstrated potent activation profiles. A microchip assay was used to show phagocytosis of the agents. Finally, the agents were tested against *S. aureus* infections *in vivo*.

I would like to highlight a few major areas of concern and some minor areas of concern:

Major:

- The septal labeling should have been more prominent in Figure 1a as these types of modifications have been performed before on vancomycin and there is almost exclusive labeling of the septa (2x higher is a low level and could simply be overlap of cell surfaces)
- It is unclear why the authors used the "N-linked" modification when this modification is known to abolish binding to D-Ala-D-Ala. This is apparently in the lack of killing of *S. aureus* cells as shown in Figure 2c. But worryingly, they see that the BODIPY variant actually appears to label the cell surface. Could this all be non-specific 'soaking' of molecules by the peptidoglycan matrix?
- Figures 1 and 2 and 4 have areas that are totally ineligible
- Why is it that growth of *S. aureus* in Figure 2 was done in a very unusual way? These should be MIC values, which represent something very different than their assay set up. As one can see, there is 10-20% remaining cell viability even at the highest concentrations. There are MIC values included in Table S1 but they left out the MIC for VISA.
- Compounds used in Figure 2 C are not the same as the ones that are used later. What is the relevance of the two sets of compounds? How do we know the MIC values of the later compounds? This is critical.
- The authors observed a decrease in efficacy of their agents against VISA but they do not return to this after Figure 2.
- It is unclear from their synthetic scheme how they can control which amino group on vancomycin is being modified starting with natural vancomycin. Usually this requires a protection/deprotection step to selectively react with individual amino acids but this does not appear to be the case or explained at all.
- Any chance that the drop off in activity of some of these agents at higher concentrations (examples in Figure 2C and Figure 3B/C/D) is related to aggregation? These peptides are extremely hydrophobic.
- Figure 4E is confusing and indirect. The authors need to show some level of killing of *S. aureus* *in vitro* as it has been done multiple times before in the literature (e.g., *Antimicrob Agents Chemother.* 2005 Aug; 49(8): 3122-8. doi: 10.1128). In the absence of this data, it is unclear if there is synergy in these agents (or even additivity).
- Why use 5 fold less of the standard concentration of vancomycin? It did not make sense and compressed the differences in the *in vivo* effect. This could be the reason of the low confidence in difference between fPEP=van and Van ($p=0.02$).

Minor:

- The number/naming of the compounds was extremely confusing.
- Any commentary on whether these agents could be active against other Gram-positive pathogens

Reviewer #2 (Remarks to the Author):

Major comments: the manuscript entitled with "Fatal attractants – Antibiotic-Chemokines Enhance the Clearance of Staphylococcus aureus Infections" presents a new concept of using antibiotic immune-therapeutics that can directly kill *S. aureus* and at the same time enhances neutrophil recruitment, achieving synergy in microbe killing. The concept of this innovative antibacterial therapy is very exciting, which is in line with the earlier published study using the dual functions achieved by the antibiotic actinonin, that directly inhibits/kills bacteria and at the same it inhibits the bacterial peptide deformylase for enhanced release of neutrophil chemoattractant, i.e., formyl peptides (PMID: 12878517). A background information that could be included as a reference.

Specific comments:

To directly demonstrate that the "antibiotic-chemokine" binds to *S. aureus* and results in an fPep gradient that stimulates neutrophil recruitment, the authors first incubated the fMLFG antibiotic-chemokine (C2) or the free fPep (fMLFG, FP1) with *S. aureus* bioparticles, then washed the samples to levels where no neutrophil recruitment was observed for the free peptide (fPep estimated to be less than 0.1 nM). Based on the data showing that the washed antibiotic-chemokine treated *S. aureus* bioparticles induced recruitment of neutrophils to the same extent as 100 nM fPep (Figure 4g), the authors conclude that both initial binding and secondary "chemokine" gradient formation following treatment with their compounds. This conclusion is very vague based on this experiment design. The authors should include the control with vancomycin alone, as there are many known and unknown chemoattractant that could be released by bacteria or from dying bacteria and killed bacterial extracts. Since a fluorescent attractant has been generated, this could be used to show that the bacterial bound material is released to form a gradient that could attract neutrophils.

#the authors consider the formylated peptides as chemokines, this is not correct, as chemoattractants (in this case) are not the same as chemokines which are a family of small cytokines. Another term should be used to describe the new chemoattractant.

#data dealing with chemical optimization of the vancomycin conjugates can be moved to supplementary fig.

#Fig 3E, F, the authors should verify which FPR (FPR1 and FPR2 in human and Fpr1 and Fpr2 in mouse) that is the preferred receptor for these modified short formyl peptides both in their human and mouse systems. This can be done with individual receptor expressing cells at least with the most promising conjugate C11. Although the authors have observed that these conjugates are active in recruiting human neutrophils, they also demonstrate a different kinetic in the recruitment for example between the conjugated fMLFG compound (FP1) and the free peptide. In addition, it is known that short formyl peptides when conjugated to other molecules, they can change binding affinity from FPR1 to FPR2 (e.g., PMID: 25447672 and PMID: 25724390). This should be commented on.

#In the discussion (starting with line 385), the authors state that they noted differences between human donors in terms of neutrophil chemotaxis and phagocytic responses to the different fPep sequences (in particular fMLFL and fMLYG). These different donor-responses could be explained by the variation in FPR number and function that are present in the human population due to polymorphisms within the FPR1 gene. These genetic variations lead to differences in response to fPep sequences, which can display similar responsiveness to fMLF, but different responses to more complex fPep sequences. It is not clear how these genetic variations lead to differences in response to fPep but not fMLF if they all are recognized by FPR1. In addition, it is well known that the fMLF response or in general FPR-mediated responses in neutrophils vary greatly from donor to donor, therefore and to make it possible to interpret the results obtained, an internal control such as fMLF should be used to normalize the responses induced by other FPR agonists. This variation might not be due to FPR1 polymorphism but rather depending on the state of neutrophils that could be affected by factors including activation state of neutrophils that can differ depending on separation and storage conditions and time to usage. The difference observed could also be due to the possibility that the complex fPep sequences bind to FPR2 (see comments above). These issues should be clarified or at least discussed.

#the authors states in line 209 that "with no high-resolution FPRs structures available", there are

indeed structures for FPR2 published (PMID: 32139677 and PMID: 32060286), that can be used for analysis.

Reviewer #3 (Remarks to the Author):

Payne et al describes a new method of overcoming antibiotic resistance through conjugating existing antibiotics to chemokines known to promote chemotaxis of neutrophils, with the intent to increase immune-mediated antibacterial activities. The manuscript demonstrates that vancomycin can be derivatized with peptides known to promote chemotaxis, and the resulting conjugates were shown to have increased antibacterial effects in the presence of neutrophils (with comparable or lower effects in the absence of neutrophils). This concept showed in vivo efficacy as well on MRSA strains compared with vancomycin alone or peptide alone. While chemokines have been studied in the past for antibacterial applications, this method is novel through conjugating with antibiotics and specifically at addressing resistance to these antibiotics.

Specific comments about the paper:

- 1) It will be helpful to address why, in some cases, high concentrations of the peptides/conjugates result in a lower level of chemotaxis. Specifically, why is there a bell-shaped curve in chemotaxis as a function of concentration of peptides/conjugates?
- 2) Related to the first point, on page 9 the authors made the point that the efficacy of the peptides should "match" the efficacy of vancomycin to avoid overstimulation. This is an important but subtle point that merits further explanation. My understanding is that if you have a really potent peptide, your efficacy on bacteria is driven by vancomycin which means that you will need a level of conjugate far higher than the potency of the peptide which means that there will be nonspecific overactivation. Is that correct?
- 3) Please double check Figure 4 on Page 15; parts of the figure legend is cut off.
- 4) I appreciate the in vivo data on the efficacy of the conjugates on mice pneumonia models exposed to MRSA, as this, in my opinion, is a key factor for publication in Nature Communications compared to a lower impact journal. However, given that this method can be applicable to even VISA strains, the authors should show in vivo efficacy with VISA side by side with MRSA, especially given the evidence that binding should be maintained or increased in VISA. After all, one can still treat MRSA with Vancomycin but that will not work for VISA.

Reviewer #4 (Remarks to the Author):

In the manuscript with the title "Fatal attractants – Antibiotic-Chemokines Enhance the Clearance of Staphylococcus aureus Infections" the authors developed antibiotic agents that act as antibiotic-chemokines, enabling augmented neutrophil recruitment to *S. aureus* in combination with direct killing. Therefore, they connected formylated peptides for neutrophil recruitment with the antimicrobial vancomycin and analyzed neutrophil chemotaxis by a Boyden Chamber in vitro assay. In addition, they used an infection-on-a-chip assay and a pneumonia mouse model, to determine if these antibiotic chemokines improve the recruitment, engulfment and killing of *S. aureus* by neutrophils.

This is an interesting idea; however, there are some crucial questions:

1. The authors perform an in vivo pneumonia assay and show that the formylated vancomycin reduce the number of cfu in the lung after 12 hours of infection, but how neutrophils contribute to this effect is not clear. The authors show that fPep and fPep-van induce chemotaxis of human neutrophils-how responds isolated mouse neutrophils towards these peptides? The authors could perform this assay using the chip assay with mouse neutrophils or a Boyden chamber assay. Since fMLF is only a very weak ligand for the mouse mFpr1, this should be analyzed in more detail.

2. Vancomycin is the last antibiotic if nothing else works- what about binding of formylated peptide to other frequently used antibiotics, which also interact with the cell envelope of *S. aureus*? The authors should show at least one example

3. If mice were treated one day after infection (and not directly) with the formylated vancomycin, how does this influence the outcome- still better than vancomycin alone?

4. Modification of formylated peptides can influence receptor preference- although they retain their chemotactic potential. Which receptor is used by formylated vancomycin? The authors should include inhibitors for FPR1 and FPR2 (e.g. tBOC, WRW4...)

5. The used *S. aureus* strains should be explained in more detail (e.g. are these Hospital-or Community-acquired MRSA strains?)

6. The discussion sounds to me weak- It should include alternative options, what is known about how our immune system could be stimulated to improve infection outcome.

7. Statistics (figure 5): since more than two groups were compared, a one way Anova should be used to compare more than two groups (instead of Mann-Whitney test).

Minor comments:

The legend of figure 4 is only partially visible, figure 4g, x-labeling lacks (time)

Page 2 line 37:Antibiotic resistance in *S. aureus* is associated with reduced levels of neutrophil recruitment, which is a vital step in triggering an immune response to resolve the infection... This is a too strong statement, since not only MRSA strains prevent neutrophil recruitment. USA300 secrete high amount of the FPR2 ligands PSMs, thereby recruiting neutrophils. The authors should weaken this statement.

We would like to thank the editorial team and the four reviewers for the time and consideration of our manuscript. Current and ongoing issues surrounding COVID-19 have led to great difficulty with this revision, however we are now very pleased to be able to include new data within this revision that – in our opinion – addresses the major questions concerning MIC of these compounds and also the nature of FPR activation by these conjugates. We have removed experiments with VISA from this revision due to major problems with experimental restrictions due to COVID and due to requirements for additional space to discuss the exciting new results we have obtained concerning FPR activation and binding. Specifically, we have added the following experimental data to the revision:

1. Inclusion of formylated peptide receptor activation data for the key conjugated formylated peptide fMChaFG and the control fMLFG. This shows different activation and binding to FPR1 & FPR2 by the different fPeps. This data is now detailed by an entirely new figure (Figure 4) and has led us to add two authors to the manuscript (Ting Fu and Chengxue Helena Qin).
2. Inclusion of the results from BODIPY labelled peptides with the attachment of the dye through the C-terminus rather than N-terminus of vancomycin. These include Airyscan images of MSSA, MRSA, *E. coli* and *B. subtilis* in the presence of the BODIPY labelled fPep, vancomycin and conjugates. These results show that the binding of vanc-fPep conjugates is specific to Gram-positive bacteria and is not mediated through the fPep. This data is included in a new panel of images in Figure 1 and new SI Figures 1 & 2 and has led us to add an author who was responsible for the synthesis of these labelled compounds (Severin Weber).
3. We have obtained MIC data for more conjugates and included this in the revision; this shows that the sequence of the fPep does not alter the antimicrobial activity of the conjugates.
4. Inclusion of mice neutrophil chemotaxis measurements in a transwell assay, which confirms that mouse neutrophils migrate to the fMChaFG conjugate also tested *in vivo*.

Specific responses to the comments from each reviewer are included below (our responses written in blue). Due to the large number of changes within this manuscript, we have not marked the individual changes.

REVIEWER COMMENTS

Reviewer #1 (Remarks to the Author):

In this manuscript, the authors describe the synthesis and application of formyl peptide covalently linked to vancomycin to operate as a dual function antibiotic-chemokine agent. The vancomycin component was intended to provide a higher local concentration of the formyl peptide, which upon activation of FPRs will result in the chemotaxis of immune cells (e.g., neutrophils). This idea appears to be fairly novel against Gram-positive pathogens and while it would likely lose its efficacy against vancomycin-resistant organisms, it should still work well with drug-sensitive pathogens. Similar concepts (antibiotic-formyl peptide conjugates) were developed against Gram-negative but were not mentioned (Antimicrob Agents Chemother. 2005 Aug;49(8):3122-8. doi: 10.1128).

Thank you for pointing out our oversight in not including this relevant research. We have now included a sentence in the introduction including this research that uses polymyxin fragment to target Gram-negative pathogens and also included the research that has used conjugates designed to target fungi.

“Aspects of linking a fPep to a targeting element has been examined for Gram-negative bacteria by using polymyxins²¹ and fungi through use of caspofungin or amphotericin B²².”

The authors first tested three points of modification as noted by V, C, and N and these correspond to three well-established points of tolerated modifications on vancomycin with a fluorophore and proceeded to evaluate chemotaxis with various formyl peptide sequences. Ultimately, they settled on a few sequences that demonstrated potent activation profiles. A microchip assay was used to show phagocytosis of the agents. Finally, the agents were tested against *S. aureus* infections in vivo. I would like to highlight a few major areas of concern and some minor areas of concern:

Major:

- The septal labeling should have been more prominent in Figure 1a as these types of modifications have been performed before on vancomycin and there is almost exclusive labeling of the septa (2x higher is a low level and could simply be overlap of cell surfaces)

To ensure we were quantifying septum binding we have added in wheat germ agglutinin staining for the cell walls as it does not bind to the septum. Therefore, only septum binding is analysed rather than overlapping cells. To further control for this we have included Hoechst nuclear staining, to ensure we are examining separating cells. This now forms the new SI Figure 1.

- It is unclear why the authors used the “N-linked” modification when this modification is known to abolish binding to D-Ala-D-Ala. This is apparently in the lack of killing of *S. aureus* cells as shown in Figure 2c. But worryingly, they see that the BODIPY variant actually appears to label the cell surface. Could this all be non-specific ‘soaking’ of molecules by the peptidoglycan matrix?

We have repeated this with newly generated C-terminal labelled BODIPY vancomycin. We repeated this work using super resolution Airyscan confocal microscopy compared to STED microscopy as it allowed us to process 100s of images in a shorter time frame whilst still distinguishing surface and septum binding (compared to 10s of images by STED). This forms the new Panel C in Figure 1. Additionally, we have added Supplementary Figure 2, which shows that no binding is observed either with the BODIPY labelled fPep or vancomycin/conjugates to *E. coli*.

- Figures 1 and 2 and 4 have areas that are totally illegible

We believe this may have been the pdf rendering and have tried to ensure that this has been fixed for this current revision.

- Why is it that growth of *S. aureus* in Figure 2 was done in a very unusual way? These should be MIC values, which represent something very different than their assay set up. As one can see, there is 10-20% remaining cell viability even at the highest concentrations.

We have added in references for this method, which is used for high throughput screening of antimicrobial activity and have altered the graphs to remove the baseline from the data which sets viability to 0 instead of the 10% that was previously used.

There are MIC values included in Table S1 but they left out the MIC for VISA.

Thank you for spotting this. We have now removed VISA from the paper to allow us to focus on FPR activation for the most interesting conjugates.

- Compounds used in Figure 2 C are not the same as the ones that are used later. What is the relevance of the two set of compounds? How do we know the MIC values of the later compounds? This is critical.

The initial set of compounds (shown originally in Figure 2c, now Figure 1e) were used to determine the optimum linkage site to vancomycin and length of PEG linker, which were C-linked compounds with no PEG linker. Using this information all further conjugates were subsequently synthesised to examine the effect of different fPep sequences, which were attached to the C-terminus of vancomycin with no PEG linker. We have determined the MIC for the main conjugated compounds fMLFG and fMChaFG along with other conjugated fPeps. Results from this work show the MIC remains the same, independent of the fPep sequence for these C-terminal linker conjugates without PEG spacers. However, for these antibiotic-chemoattractants we are using them well below their MIC and the critical factor for the antibiotic portion is that it binds to the bacteria, not that it has direct antimicrobial activity.

- The authors observed a decrease in efficacy of their agents against VISA but they do not return to this after Figure 2.

We have added additional data (FPR interaction) to the paper and in doing so found that VISA was somewhat removed from the focus of our manuscript; due to additional data we also no longer have the space to include discussion of VISA. However, in designing this conjugate we

selected vancomycin as the targeting element as although there is a loss in antimicrobial activity against VISA strains, the binding of vancomycin remains high to the cell wall. Thus, our conjugates would still be capable of delivery of fPeps to the surface of these bacteria.

- It is unclear from their synthetic scheme how they can control which amino group on vancomycin is being modified starting with natural vancomycin. Usually this requires a protection/deprotection step to selectively react with individual amino acids but this does not appear to be the case or explained at all.

We have found it more efficient to not include a protection/deprotection step. In this case, both amino groups on the vancomycin will be targeted, creating regioisomers that are then able to be separated chromatographically. We validated the structure of each compound of interest by performing a small-scale test hydrolysing the sugars from both compounds, thus revealing the linkage site and hence identity of each of the two compounds.

- Any chance that the drop off in activity of some of these agents at higher concentrations (examples in Figure 2C and Figure 3B/C/D) is related to aggregation? These peptides are extremely hydrophobic.

High concentrations of fPep will inhibit chemotaxis: typical concentration response curves are bell-shaped for chemotaxis activity, which is what is observed here. These compounds were stable in solution, with no signs of turbidity at higher concentrations. Aliquots were prepared fresh in DMSO before each experiment, with final concentrations in the nanomolar range thus reducing aggregation potential. Additionally, our strategy in conjugating these fPeps to vancomycin takes advantage of the highly polar nature of this antibiotic, which allowed us to explore a range of fPep sequences with different hydrophobicity and to quantify their effects on FPR activation.

- Figure 4E is confusing and indirect. The authors need to show some level of killing of *S. aureus* in vitro as it has been done multiple times before in the literature (e.g., *Antimicrob Agents Chemother.* 2005 Aug;49(8):3122-8. doi: 10.1128). In the absence of this data, it is unclear if there is synergy in these agents (or even additivity).

Microfluidics allowed us to combine monitoring of chemotaxis, phagocytosis, and *S. aureus* growth in real time. This makes it a most insightful assay as it allows the coupling of chemotaxis and bacterial growth inhibition to be observed in real time. The assay suggested in *Antimicrob Agents Chemother.* 2005 Aug;49(8):3122-8 instead relies on the mixing of immune cells with bacteria and counting CFUs at an end point, which is not able to assay all of these features.

- Why use 5 fold less of the standard concentration of vancomycin? It did not make sense and compressed the differences in the in vivo effect. This could be the reason of the low confidence in difference between fPEP=van and Van ($p=0.02$).

The vancomycin moiety in our conjugates is being used for recruitment of the fPep to bacteria, not for direct antimicrobial killing. Hence, we utilise our conjugates below the

standard concentration of vancomycin to be able to avoid the potential for resistance to emerge to the vancomycin targeting element of our conjugates.

Minor:

- The number/naming of the compounds was extremely confusing.

We have changed the numbering to align the conjugated and fPep numbering, e.g. fMLFG in the free and conjugate forms are FP1 and C1 respectively while fMChaFG in the free and conjugate forms are FP11 and C11, respectively. The BODIPY labelled compounds are all labelled with a B#, while all free fPeptides are shown as FP# to help make the manuscript easier to follow. To help distinguish between the different linkage strategies between the conjugates all N-linked conjugates are labelled as N#, C-linked conjugates as C# and V-linked conjugates as V# (see Figure SI 8 and page 7, line 6-7).

- Any commentary on whether these agents could be active against other Gram-positive pathogens

We have added in binding of BODIPY labelled conjugate to *B. subtilis* in Figure 1. We have also added in statement on use of these conjugates against other Gram-positive bacteria on page 6, line 31-33. The generality of vancomycin binding to the cell wall of Gram-positive bacteria certainly indicates that broader activity is possible, which is an aspect of this work that will be explored in subsequent investigations.

Reviewer #2 (Remarks to the Author):

Major comments: the manuscript entitled with “Fatal attractants – Antibiotic-Chemokines Enhance the Clearance of Staphylococcus aureus Infections” presents a new concept of using antibiotic immune-therapeutics that can directly kill *S. aureus* and at the same time enhances neutrophil recruitment, achieving synergy in microbe killing. The concept of this innovative antibacterial therapy is very exciting, which is in line with the earlier published study using the dual functions achieved by the antibiotic actinonin, that directly inhibits/kills bacteria and at the same it inhibits the bacterial peptide deformylase for enhanced release of neutrophil chemoattractant, i.e., formyl peptides (PMID: 12878517). A background information that could be included as a reference.

Thank you, we have added this into to the introduction on page 3, line 24-26.

Specific comments:

To directly demonstrate that the “antibiotic-chemokine” binds to *S. aureus* and results in an fPep gradient that stimulates neutrophil recruitment, the authors first incubated the fMLFG antibiotic- chemokine (C2) or the free fPep (fMLFG, FP1) with *S. aureus* bioparticles, then washed the samples to levels where no neutrophil recruitment was observed for the free peptide (fPep estimated to be less than 0.1 nM). Based on the data showing that the washed antibiotic-chemokine treated *S. aureus* bioparticles induced recruitment of neutrophils to the same extent as 100 nM fPep (Figure 4g), the authors conclude that both initial binding and secondary “chemokine” gradient formation following treatment with their compounds. This conclusion is very vague based on this experiment design. The authors should include the control with vancomycin alone, as there are many known and unknown chemoattractant that could be released by bacteria or from dying bacteria and killed bacterial extracts.

This is an excellent point. We have used a vancomycin only control in this experiment and we see no recruitment of neutrophils to the vancomycin alone (SI Figure 4) or any significantly improved recruitment to *S. aureus* in the presence of vancomycin compared to *S. aureus* alone (see additional Review Figure 1 below). This situation may be altered if the concentration of vancomycin is increased to the point that it results in killing of *S. aureus*, which as suggested could lead to the release of chemoattractants. However, here we do not see this at the nanomolar concentration we are employing our compounds, which did not directly inhibit *S. aureus* growth (SI Figure 5). We show the results of the vancomycin alone treatment in Supplementary Figure 4 for the transwell assay and below for the microfluidics assay, both of which indicate that vancomycin itself has no chemoattractant properties for neutrophils at the concentrations tested.

Review Figure 1. Recruitment of neutrophils to *S. aureus* in presence of vancomycin

The recruitment of neutrophils to *S. aureus* over time was tested in the presence of different compounds at 100 nM using our microfluidics assay. The fMLFG peptide both free and in its conjugated form improves recruitment of neutrophils to *S. aureus*. Compounds without formylated peptides (vancomycin (van), MLFG in free and conjugated forms (MLFG and MLFG=van) resulted in the similar recruitment of neutrophils as the *S. aureus* alone (DMSO + *S. aureus*). In the absence of *S. aureus* these compounds resulted in no recruitment. Data shown from a single donor.

Since a fluorescent attractant has been generated, this could be used to show that the bacterial bound material is released to form a gradient that could attract neutrophils.

We have attempted to quantify this. However, we were able to visualise the gradient over time only when using unwashed samples. When using washed samples, we see binding to the bacterial surface using high resolution imaging (Figure 1). However, the gradient that forms is not visually evident or measurable due to the high fluorescence on the bacterial surface versus the surrounding area. We can only indirectly report the presence of the gradient of compound forming around the washed bacteria thanks to the observation of chemotaxis of neutrophils in these microfluidic devices. If the conjugates remained very tightly bound to the bacteria, then the recruitment of neutrophils would be anticipated to be the same as if the sample was bacteria alone. Given that we see enhanced chemotaxis in our treated samples, we infer that this gradient is indeed being formed.

#the authors consider the formylated peptides as chemokines, this is not correct, as chemoattractants (in this case) are not the same as chemokines which are a family of small cytokines. Another term should be used to describe the new chemoattractant.

Thank you for correcting this error on our part – we have changed the naming to chemoattractant throughout the manuscript.

#data dealing with chemical optimization of the vancomycin conjugates can be moved to supplementary fig.

We have kept this in the main text as we believe it is important in understanding how to generate these antibiotic-chemoattractants, particularly in light of comments from other reviewers.

#Fig 3E, F, the authors should verify which FPR (FPR1 and FPR2 in human and Fpr1 and Fpr2 in mouse) that is the preferred receptor for these modified short formyl peptides both in their human and mouse systems. This can be done with individual receptor expressing cells at least with the most promising conjugate C11.

Thank you for this suggestion! We appreciate this point and indeed feel that this was a major unanswered question. Thus, we have directed significant effort together with collaborators to be able to address this and to be able to now include the data in this revision. We have added data concerning human FPR1 and 2 receptor binding both by a competition binding assay and downstream signalling (ERK and calcium mobilisation) using CHO cells overexpressing human FPR1 or FPR2 (Figure 4) as suggested. Due to this increased focus on FPR activation, we have thus removed discussion of VISA from the manuscript to be able to address these new results in appropriate detail.

Although the authors have observed that these conjugates are active in recruiting human neutrophils, they also demonstrate a different kinetic in the recruitment for example between the conjugated fMLFG compound (FP1) and the free peptide. In addition, it is known that short formyl peptides when conjugated to other molecules, they can change binding affinity from FPR1 to FPR2 (e.g., PMID: 25447672 and PMID: 25724390). This should be commented on.

We have added in further discussion regarding FPR1 & FPR2 interaction into both the results (page 15, line 25-31) and discussion (page 21, paragraph commencing line 21). This data does support the hypothesis that conjugation changes the preference of interaction from FPR1 to FPR2 for these fPep conjugates.

#In the discussion (starting with line 385), the authors state that they noted differences between human donors in terms of neutrophil chemotaxis and phagocytic responses to the different fPep sequences (in particular fMLFL and fMLYG). These different donor-responses could be explained by the variation in FPR number and function that are present in the human population due to polymorphisms within the FPR1 gene. These genetic variations lead to differences in response to fPep sequences, which can display similar responsiveness to fMLF, but different responses to more complex fPep sequences. It is not clear how these genetic variations lead to differences in response to fPep but not fMLF if they all are recognized by FPR1. In addition, it is well known that the fMLF response or in general FPR-mediated responses in neutrophils vary greatly from donor to donor, therefore and to make it possible to interpret the results obtained, an internal control such as fMLF should be used to normalize the responses induced by other FPR agonists.

We agree, which is why we have used fMLFG as our internal control to normalise responses. In this regard the chemotaxis observed at 100 nM fMLFG is set as 100% for each donor. We have highlighted this in each of the figure legends by stating "Chemotaxis was calculated relative

to the no protein control and 100% chemotaxis set as the neutrophil recruitment observed to fMLFG at 100 nM for each donor". We have also outlined the reasoning behind calculating relative chemotaxis due to differences between donors in the methods on Line 27 on page 27. We have used fMLFG rather than fMLF to normalise responses as our library of fPeps was based upon this sequence of four amino acids. We have also added fMLF as the comparator peptide in the CHO cells overexpressing human FPR1 or FPR2 (Figure 4).

This variation might not be due to FPR1 polymorphism but rather depending on the state of neutrophils that could be affected by factors including activation state of neutrophils that can differ depending on separation and storage conditions and time to usage.

To minimise this possibility, we use a standardised method for isolating the neutrophils. We also isolate neutrophils directly from blood, purify and then test them in our assays within 12 h, reducing all unnecessary handling to minimise activation before use. Controls are always used for each experiment to ensure that neutrophils have not been pre-activated due to inappropriate handling.

The difference observed could also be due to the possibility that the complex fPep sequences bind to FPR2 (see comments above). These issues should be clarified or at least discussed.

Absolutely – we have added in additional data regarding FPR activation (Figure 4) as outlined above. Both the results (page 15, line 25-31) and discussion (page 21, paragraph commencing line 21) now contain further information regarding the conjugate and free fPep interactions with FPR1 & FPR2, which we believe is a major improvement in this revised version.

#the authors states in line 209 that "with no high-resolution FPRs structures available", there are indeed structures for FPR2 published (PMID: 32139677 and PMID: 32060286), that can be used for analysis.

Thank you for pointing this out – we have added these references along with further comments on the binding pocket of FPR2 and interaction with our larger conjugate compounds in the discussion and results. And – briefly to clarify – when we initiated our studies these structures were not available.

Reviewer #3 (Remarks to the Author):

Payne et al describes a new method of overcoming antibiotic resistance through conjugating existing antibiotics to chemokines known to promote chemotaxis of neutrophils, with the intent to increase immune-mediated antibacterial activities. The manuscript demonstrates that vancomycin can be derivatized with peptides known to promote chemotaxis, and the resulting conjugates were shown to have increased antibacterial effects in the presence of neutrophils (with comparable or lower effects in the absence of neutrophils). This concept showed in vivo efficacy as well on MRSA strains compared with vancomycin alone or peptide alone. While chemokines have been studied in the past for antibacterial applications, this method is novel through conjugating with antibiotics and specifically at addressing resistance to these antibiotics.

Specific comments about the paper:

1) It will be helpful to address why, in some cases, high concentrations of the peptides/conjugates result in a lower level of chemotaxis. Specifically, why is there a bell-shaped curve in chemotaxis as a function of concentration of peptides/conjugates?

High concentrations of a chemoattract or chemokine will inhibit chemotaxis; this is due to the requirement for the immune cells to cease movement once they have encountered their bacterial target. This results in the concentration response curve being bell shaped for chemotaxis activity, which is a well know phenomenon that we acknowledge was not appropriately discussed in the original manuscript.

2) Related to the first point, on page 9 the authors made the point that the efficacy of the peptides should "match" the efficacy of vancomycin to avoid overstimulation. This is an important but subtle point that merits further explanation. My understanding is that if you have a really potent peptide, your efficacy on bacteria is driven by vancomycin which means that you will need a level of conjugate far higher than the potency of the peptide which means that there will be nonspecific overactivation. Is that correct?

Our use of language here was possibly unclear; rather than ensure that the affinity of vancomycin and the activity maximum of each peptide was the same, we rather wanted to ensure that we would have access to a range of different chemotaxis profiles to explore for our conjugates. All our conjugates all have a shift towards higher concentrations in their bell curves (right) which helps to ensure that these are not overly stimulating neutrophils inappropriately. We have re-worded this sentence to clarify this point.

3) Please double check Figure 4 on Page 15; parts of the figure legend is cut off.

Apologies for this, which appeared to be a problem when we converted the document to pdf. We have endeavoured to correct this in the revision.

4) I appreciate the in vivo data on the efficacy of the conjugates on mice pneumonia models exposed to MRSA, as this, in my opinion, is a key factor for publication in Nature Communications compared to a lower impact journal. However, given that this method can

be applicable to even VISA strains, the authors should show in vivo efficacy with VISA side by side with MRSA, especially given the evidence that binding should be maintained or increased in VISA. After all, one can still treat MRSA with Vancomycin but that will not work for VISA.

Unfortunately, we do not have ethics approval for this assay. Additionally, COVID-19 distancing restrictions enforced during the period of the revision (and indeed ongoing) have meant that establishing new animal models in mice has been impossible, whilst work supported by existing models and ethics approvals has been extremely challenging to perform due to needing two people to perform this assay and government requirements of physical distancing and room occupancy. Given these difficulties and the need to include our exciting results concerning the alterations in FPR binding/ signalling seen with our conjugates, we have removed analysis of our compounds with VISA from this revision.

Reviewer #4 (Remarks to the Author):

In the manuscript with the title "Fatal attractants – Antibiotic-Chemokines Enhance the Clearance of Staphylococcus aureus Infections" the authors developed antibiotic agents that act as antibiotic-chemokines, enabling augmented neutrophil recruitment to S. aureus in combination with direct killing. Therefore, they connected formylated peptides for neutrophil recruitment with the antimicrobial vancomycin and analyzed neutrophil chemotaxis by a Boyden Chamber in vitro assay. In addition, they used an infection-on-a-chip assay and a pneumonia mouse model, to determine if these antibiotic chemokines improve the recruitment, engulfment and killing of S. aureus by neutrophils.

This is an interesting idea; however, there are some crucial questions:

1. The authors perform an in vivo pneumonia assay and show that the formylated vancomycin reduce the number of cfu in the lung after 12 hours of infection, but how neutrophils contribute to this effect is not clear. The authors show that fPep and fPep-van induce chemotaxis of human neutrophils-how responds isolated mouse neutrophils towards these peptides? The authors could perform this assay using the chip assay with mouse neutrophils or a Boyden chamber assay. Since fMLF is only a very weak ligand for the mouse mFpr1, this should be analyzed in more detail.

We have examined the fMLFG and fMChaFG in both free and conjugate form for the recruitment of mouse neutrophils in the transwell assay. Both resulted in the recruitment of mouse neutrophils, with fMChaFG resulting in greater recruitment than fMLFG.

2. Vancomycin is the last antibiotic if nothing else works- what about binding of formylated peptide to other frequently used antibiotics, which also interact with the cell envelope of S. aureus? The authors should show at least one example

This is possible, but beyond the scope of this manuscript. We refer the reviewer to work that has successfully attached fPep to polymyxins for activity against Gram-negative bacteria and caspofungin for targeting fungi. However, these works did not optimise the fPep sequence for the payload, whilst we have not only performed this optimisation but have now also interrogated the causes for the improved properties of our optimised fPep sequences.

3. If mice were treated one day after infection (and not directly) with the formylated vancomycin, how does this influence the outcome- still better than vancomycin alone?

Unfortunately, we do not have ethics approval for this assay. As was the case for our answer for point (4) from the previous reviewer, COVID-19 distancing has meant animal work has been extremely challenging to perform due to government requirements of physical distancing and room occupancy.

4. Modification of formylated peptides can influence receptor preference- although they retain their chemotactic potential. Which receptor is used by formylated vancomycin? The authors should include inhibitors for FPR1 and FPR2 (e.g. tBOC, WRW4...)

Thank you for this suggestion! We have added in new experiments to examine the interaction of the two key fPeps and conjugates with FPRs, using a competition binding assay between these compounds and a fluorescent rhodamine labelled FPR2 agonist. Additionally, we have examined the downstream activation of the FPRs by using CHO cells overexpressing either human FPR1 or FPR2 and examining the phosphorylation of ERK and calcium mobilisation. This work has formed the new Figure 4 and additional paragraph in the discussion starting line 21, page 21.

5. The used *S. aureus* strains should be explained in more detail (e.g. are these Hospital- or Community-acquired MRSA strains?)

Thank you for pointing out our oversight in not including these details. These are hospital acquired strains and this information has been added to SI Table 1.

6. The discussion sounds to me weak- It should include alternative options, what is known about how our immune system could be stimulated to improve infection outcome.

We have added additional discussion on FPR activation and signalling, which we believe now significantly strengthens the discussion.

7. Statistics (figure 5): since more than two groups were compared, a one way Anova should be used to compare more than two groups (instead of Mann-Whitney test).

Thank you for raising this point. We have changed this to a one-way ANOVA.

Minor comments:

The legend of figure 4 is only partially visible, figure 4g, x-labeling lacks (time)

Apologies for this, which was a pdf rendering problem. We believe that this error has been fixed with this revision.

Page 2 line 37:Antibiotic resistance in *S. aureus* is associated with reduced levels of neutrophil recruitment, which is a vital step in triggering an immune response to resolve the infection...This is a too strong statement, since not only MRSA strains prevent neutrophil recruitment. USA300 secrete high amount of the FPR2 ligands PSMs, thereby recruiting neutrophils. The authors should weaken this statement.

This is a very good point. We have altered the original statement, which now reads- "Avoiding detection by cells of our innate immune response ensures that first line responders such as neutrophils cannot come into play to protect us from this infection⁴. Additionally, resistance to clinical antibiotics in *S. aureus* is leading to the emergence of strains with greater immune evasion^{5, 6}. The increased difficulty in eradicating these strains therefore leads to persistent and deadly infections⁶."

REVIEWER COMMENTS

Reviewer #3 (Remarks to the Author):

Many thanks, I am satisfied with the authors' responses to my comments.

[Reviewer #1 was not able to assess this version of the manuscript. Upon the editors' request, reviewer #3 kindly assessed the authors' responses to reviewer #1 and provided the following additional comments]

There are a few points previously raised by reviewer #1 that I felt were not adequately addressed:

- Point #2 ("It is unclear why the authors used the "N-linked" modification..."):

Reviewer #3: Not sure if the authors really addressed this, especially given that the C-linked analogues did show activity. Also, not sure if the E. coli experiment is an appropriate control since gram negative bacteria have a smaller peptidoglycan matrix. The authors should consider a competition experiment with unlabeled Vancomycin to outcompete the binding of the labeled analogues.

- Point #6 ("Compounds used in Figure 2 C are not the same as the ones that are used later"):

Reviewer #3: I appreciate that the binding and not the MIC is the most critical, but it would still be helpful to show the data that shows the MICs are the same, even if in the supporting figure.

- Point #8 ("It is unclear from their synthetic scheme how they can control..."):

Reviewer #3: A discussion on this in the supporting information would be helpful. Also, please provide the method for purification of the regioisomers.

- Point #10 ("Figure 4E is confusing and indirect"):

Reviewer #3: I agree with the author that their method is superior to the method in the Antimicrob Agent Chemother, but I also see the point of the reviewer because at this point there still isn't a direct head to head comparison with S. aureus growth w/ or w/o the effect of immune recognition. Can the authors repeat this experiment using the same method, but with multiple concentrations, and in the presence and absence of neutrophils?

Reviewer #4 (Remarks to the Author):

Although it is still not 100 % clear through which mouse Formyl-peptide receptor the conjugated formylated peptides activate mouse neutrophils, it is clear now that these conjugates induce migration. All questions are sufficient answered, additional experiments were conducted. This is an interesting approach which should be considered in the future for a more efficient treatment of severe MRSA infections.

Response to reviewers:

- Point #2 ("It is unclear why the authors used the "N-linked" modification..."):

Reviewer #3: Not sure if the authors really addressed this, especially given that the C-linked analogues did show activity. Also, not sure if the *E. coli* experiment is an appropriate control since gram negative bacteria have a smaller peptidoglycan matrix. The authors should consider a competition experiment with unlabeled Vancomycin to outcompete the binding of the labeled analogues.

We agree with reviewer 1 that the N-linked modification did not fit with the rest of the paper where we have continued with the C-linked version. We therefore generated the C-linked BODIPY labelled version for the first revision of this manuscript and thus removed all the N-linked BODIPY labelled work from the paper. We apologise that this was not clearly indicated in the previous response to reviewer's comments.

We believe that the inclusion of binding studies with Gram-negative *E. coli* is important as it shows that our compounds are specifically interacting with the exposed cell wall of *S. aureus* (and other Gram-positive bacteria including *B. subtilis*) rather than non-selectively with bacteria in general. This specific binding observed, combined with the observation that fPep alone and Bodipy alone did not interact at this same level to the bacterial cell surface, suggests that we are not observing non-specific soaking by the peptidoglycan matrix.

- Point #6 ("Compounds used in Figure 2 C are not the same as the ones that are used later"):

Reviewer #3: I appreciate that the binding and not the MIC is the most critical, but it would still be helpful to show the data that shows the MICs are the same, even if in the supporting figure.

We agree with reviewer 1 and 3 that it was important to compare the MIC values of the different conjugates. Our apologies – we did not state in our original response that this MIC data is now included in the manuscript and is discussed in text on page 10 line 9: "The conjugates **C18-19**, along with fMLFG=van (**C1**) and fMChaFG=van (**C11**) were all found to have MIC values of 1.8 μ M against both MRSA and MSSA strains tested. This indicates that the sequence of the formylated peptide attached to vancomycin does not alter antimicrobial activity, unlike that seen with varying the attachment site of the fPep to vancomycin." To help clarify this further we have also added the line (on page 10 line 13)- "It also allows the direct comparison of the neutrophil killing activity of these different conjugates as they possess the same level of direct antimicrobial activity against the bacterial strains tested."

- Point #8 ("It is unclear from their synthetic scheme how they can control..."):

Reviewer #3: A discussion on this in the supporting information would be helpful. Also, please provide the method for purification of the regioisomers.

Thank you for pointing out this oversight. We have added the following description to the methods on page 24 line 16: “N-terminal and vancosamine conjugation reactions were performed in one pot without protection; the resultant regioisomers were separated using RP-HPLC, with the identity of each compound confirmed by small-scale hydrolysis. The resultant mass difference caused by the loss of either vancosamine or the modified vancosamine thus allowed the identity of each compound to be confirmed.”

We have also added the following text to the caption of Figure SI 6: “The identities of the regioisomers **GPA3** and **4** were confirmed by comparing the LCMS analysis after acidolysis with TFA. In the case of **GPA3**, acidolysis led to the characterization of the vancomycin aglycone bearing the azidopentyl moiety at the N-terminus (MW = 1267.4 Da), while **GPA4** acidolysis led to formation of the vancomycin aglycone (MW = 1142.4 Da).”

- Point #10 ("Figure 4E is confusing and indirect"):

Reviewer #3: I agree with the author **that their method is superior** to the method in the Antimicrob Agent Chemother, but I also see the point of the reviewer because at this point there still isn't a direct head to head comparison with *S. aureus* growth w/ or w/o the effect of immune recognition. Can the authors repeat this experiment using the same method, but with multiple concentrations, and in the presence and absence of neutrophils?

We are pleased that the reviewer agrees with our assessment of the advantages of our microfluidics method for assessing the activity of our compounds, which is particularly powerful as it mimics and the neutrophil process of recruitment through to phagocytosis that occurs to combat infections (see Figure below). The microfluidics experiments thus allow us to monitor this entire process in real time to determine the effect of the neutrophil response on *S. aureus* growth.

Reviewer Response Figure 1. Comparison of the process of neutrophil recruitment in response to infection in humans and in microfluidic devices. Neutrophils (blue) patrolling the area are activated and recruited to the site of infection by following a gradient of chemoattracts (orange). At the site of infection *S. aureus* (yellow) are phagocytosed by neutrophils during the process of bacterial killing by the neutrophils.

In response to this comment, we would like to point out that we have included the head-to-head comparison of *S. aureus* growth in the presence and absence of neutrophil recruitment in **Figure 3** and in the **Figure SI 5**. **Figure 3c & e** shows that with no recruitment of neutrophils (in this case when no compound is present) the growth of *S. aureus* increases exponentially with time. This growth is curbed if our conjugates are present and neutrophils subsequently recruited. The conjugates themselves are not present in a high enough concentration to have a direct antimicrobial effect on *S. aureus* and therefore the inhibition of *S. aureus* growth can be ascribed directly to the activity of neutrophils. This is further confirmed in **Figure SI 5** where the growth of *S. aureus* was shown to be the same either in the presence or absence of the

MLFG-vancomycin conjugate (1000 nM). Given that there is no difference in the direct antimicrobial activity of this conjugate compared to either the fMChaFG or fMLFG conjugates, the lack of growth inhibition seen here is due to the lack of neutrophil recruitment.

We have added to the text on page 12 line 27 to help clarify and further explain these results. “The control of *S. aureus* growth here is due to the neutrophil action, rather than the direct antimicrobial activity of the compounds: this was confirmed by the observations that the MLFG=van compound, which does not result in neutrophil recruitment (unlike **C1** & **C11**) whilst displaying similar direct antimicrobial activity as **C1** and **C11** resulted in no observable growth inhibition of *S. aureus* (**Supplementary Figure 5**).

REVIEWERS' COMMENTS

Reviewer #3 (Remarks to the Author):

Many thanks for the revisions and clarifications. I believe this manuscript is much improved and now ready for publication.